# Estrogen mediates sex differences in preoptic neuropeptide and pituitary hormone production in medaka

Junpei Yamashita[1], Yuji Nishiike[1], Thomas Fleming[1], Daichi Kayo[1] & Kataaki Okubo [1✉]

The preoptic area (POA) is one of the most evolutionarily conserved regions of the vertebrate brain and contains subsets of neuropeptide-expressing neurons. Here we found in the teleost medaka that two neuropeptides belonging to the secretin family, pituitary adenylate cyclase-activating polypeptide (Pacap) and vasoactive intestinal peptide (Vip), exhibit opposite patterns of sexually dimorphic expression in the same population of POA neurons that project to the anterior pituitary: Pacap is male-biased, whereas Vip is female-biased. Estrogen secreted by the ovary in adulthood was found to attenuate Pacap expression and, conversely, stimulate Vip expression in the female POA, thereby establishing and maintaining their opposite sexual dimorphism. Pituitary organ culture experiments demonstrated that both Pacap and Vip can markedly alter the expression of various anterior pituitary hormones. Collectively, these findings show that males and females use alternative preoptic neuropeptides to regulate anterior pituitary hormones as a result of their different estrogen milieu.

[1] Department of Aquatic Bioscience, Graduate School of Agricultural and Life Sciences, The University of Tokyo, Bunkyo, Tokyo, Japan.
✉email: okubo@marine.fs.a.u-tokyo.ac.jp

The preoptic area (POA) is an evolutionarily ancient brain region that is hodologically, molecularly and functionally conserved across vertebrates[1]. It contains distinct populations of neuropeptide-expressing neurons that mediate a wide array of behavioral and homeostatic functions, including social behaviors and pituitary hormone secretion[2–5]. These neurons receive input from multiple sensory modalities and express sex steroid receptors, thereby integrating sensory and steroidal information for the control of behavioral and neuroendocrine functions. Another distinguishing feature of the POA is the presence of prominent differences, from the anatomical to the molecular level, between the sexes[6, 7]. Probably the best-known example is in rodents, where the sexually dimorphic nucleus of the POA, a small cluster of neurons within the central subdivision of the POA, is 3–5 times larger in males[8]. Nevertheless, this neuronal cluster has not been clearly tied to any specific behavioral or physiological trait, although several studies suggest that it has a role in sexual orientation and mating behavior[9]. As such, the functional importance of sex differences in the POA remains largely undetermined.

Pituitary adenylate cyclase-activating polypeptide (PACAP) is a member of the secretin family of neuropeptides that was initially identified by its ability to stimulate adenylate cyclase activity in anterior pituitary cells[10]. It has since been implicated in a myriad of behavioral and physiological functions, including circadian rhythms, learning and memory, anxiety and responses to stress and brain injury[11, 12]. Within the secretin family, PACAP is most closely related in structure to vasoactive intestinal peptide (VIP), which was originally identified as a vasodilator in the gastrointestinal tract but has since been recognized as a neuropeptide[13]. PACAP and VIP share common G-protein-coupled receptors, designated ADCYAP1 receptor 1 (ADCYAP1R1), VIP receptor 1 (VIPR1) and VIP receptor 2 (VIPR2) (also known as PAC1, VPAC1 and VPAC2, respectively) and thus have overlapping functions[11, 12].

In recent years, the teleost fish medaka (*Oryzias latipes*) has emerged as a valuable model organism for studying brain sex differences owing to, for example, a short generation time, robust sex differences in behavioral and physiological phenotypes and ease of genetic and experimental manipulations including those that affect sexual phenotypes[14–17]. In the present study, we searched for genes that are differentially expressed by sex in the medaka brain and found that the *adcyap1* gene encoding Pacap exhibits extreme male-biased expression in the POA. Given the structural and functional similarities and shared receptor usage of PACAP and VIP, we also examined expression of the *vip* gene encoding Vip in the medaka brain, which showed that, in contrast to *adcyap1*, *vip* exhibits female-biased expression in the POA. Our study further demonstrated that, as a result of their different estrogen milieu, males and females primarily utilize Pacap and Vip, respectively, to regulate the production of anterior pituitary hormones.

## Results

***adcyap1* and *vip* exhibit opposite patterns of sex-biased expression in the medaka brain**. We searched for genes that are differentially expressed between the sexes in the medaka brain by suppression subtractive hybridization[18]. A cDNA fragment derived exclusively from the male-enriched library had best BLAST hits to *adcyap1* in other species. Screening of a full-length medaka brain cDNA library facilitated the isolation of a 2883-bp cDNA (excluding the poly(A)$^+$ tail) corresponding to this male-enriched fragment (deposited in GenBank with accession number LC579549). Phylogenetic tree analysis confirmed that the cDNA encoded the medaka ortholog of known PACAP precursors in

other species and thus represented medaka *adcyap1* (Fig. 1a). Identification of *adcyap1* as differentially expressed between the sexes prompted us to investigate whether its close relative *vip* showed a similar sex difference. We isolated *vip* by rapid amplification of cDNA ends (RACE) on medaka brain poly(A)$^+$ RNA, which facilitated the isolation of a 1795-bp cDNA (excluding the poly(A)$^+$ tail) encoding a putative Vip precursor (deposited in GenBank with accession number LC579550). Phylogenetic tree analysis confirmed the identity of this cDNA as representing medaka *vip* (Fig. 1a).

Differential expression of *adcyap1* by sex was verified by real-time PCR, which showed that the male brain had 1.45-fold higher levels of overall *adcyap1* expression relative to the female brain ($p$ < 0.0001) (Fig. 1b). In contrast and unexpectedly, examination of overall brain *vip* expression revealed that it was 1.43-fold higher in females than in males ($p = 0.0007$) (Fig. 1b). Northern blot analysis detected 3.2 and 2.2 kb transcripts for *adcyap1*, both of which were more abundant in the male than in the female brain (Fig. 1c). This result further verified the male bias in *adcyap1* expression and indicated that the isolated *adcyap1* cDNA was full length and corresponded to the longer transcript (the cDNA corresponding to the shorter transcript, arising from alternative use of polyadenylation signals, was also isolated). Likewise, the detection of a 2.0 kb transcript for *vip*, which was more abundant in the female brain, verified the female bias in *vip* expression and indicated that the isolated *vip* cDNA was full length (Fig. 1c).

Given that medaka are diurnal fish and undergo daily cycles of gametogenesis and spawning, we assessed diurnal variation in overall *adcyap1* and *vip* expression levels in the brain by real-time PCR. *adcyap1* was consistently expressed at significantly higher levels in the male than in the female brain throughout the day, with the exception of the late light phase (main effect of sex, $p$ < 0.0001; main effect of time, $p$ < 0.0001; interaction between sex and time, $p$ < 0.0001; sex difference at each timepoint, $p$ < 0.0001, < 0.0001, = 0.0002, 0.0694, 0.0099 and 0.0280 at 6:00, 10:00, 14:00, 18:00, 22:00 and 2:00, respectively) (Fig. 1d). The expression of *vip* was significantly higher in the female than in the male brain at the majority of time points assessed, except during the late light phase (main effect of sex, $p$ < 0.0001; main effect of time, $p$ < 0.0001; interaction between sex and time, $p = 0.0233$; sex difference at each timepoint, $p = 0.0017$, 0.0030, 0.0011, 0.9017, > 0.9999 and 0.0001 at 6:00, 10:00, 14:00, 18:00, 22:00 and 2:00, respectively) (Fig. 1d).

We also explored when differential expression of *adcyap1* and *vip* by sex is established during growth and sexual maturation. Examination of brains at different ages by real-time PCR revealed that the sex bias in both *adcyap1* and *vip* expression increased steadily with age (for both *adcyap1* and *vip*: main effect of sex, $p$ < 0.0001; main effect of age, $p$ < 0.0001; interaction between sex and age, $p$ < 0.0001) (Fig. 1e). Significant sex bias in *adcyap1* and *vip* expression was evident as early as, respectively, 3 months of age ($p = 0.0001$), when fish had become sexually mature and spawned, and 2 months of age ($p = 0.0200$), when secondary sexual characteristics were well developed but spawning had not yet occurred (Fig. 1e).

**The POA is the major source of the sex-biased expression of *adcyap1* and *vip* in the brain**. We investigated the source of the sex-biased expression of *adcyap1* and *vip* in medaka brain by using quantitative in situ hybridization. Neurons expressing *adcyap1* were identified in various brain nuclei, including the following: GL in the olfactory bulb; Dl/Dm/Dd/Dp in the dorsal telencephalon; Vv and Vd/Vs/Vp in the ventral telencephalon; PMp, PPa and SC in the POA; VM, VL and DP/CP in the thalamus; rHd and lHd in the habenula; NC in the pretectum; NPPv,

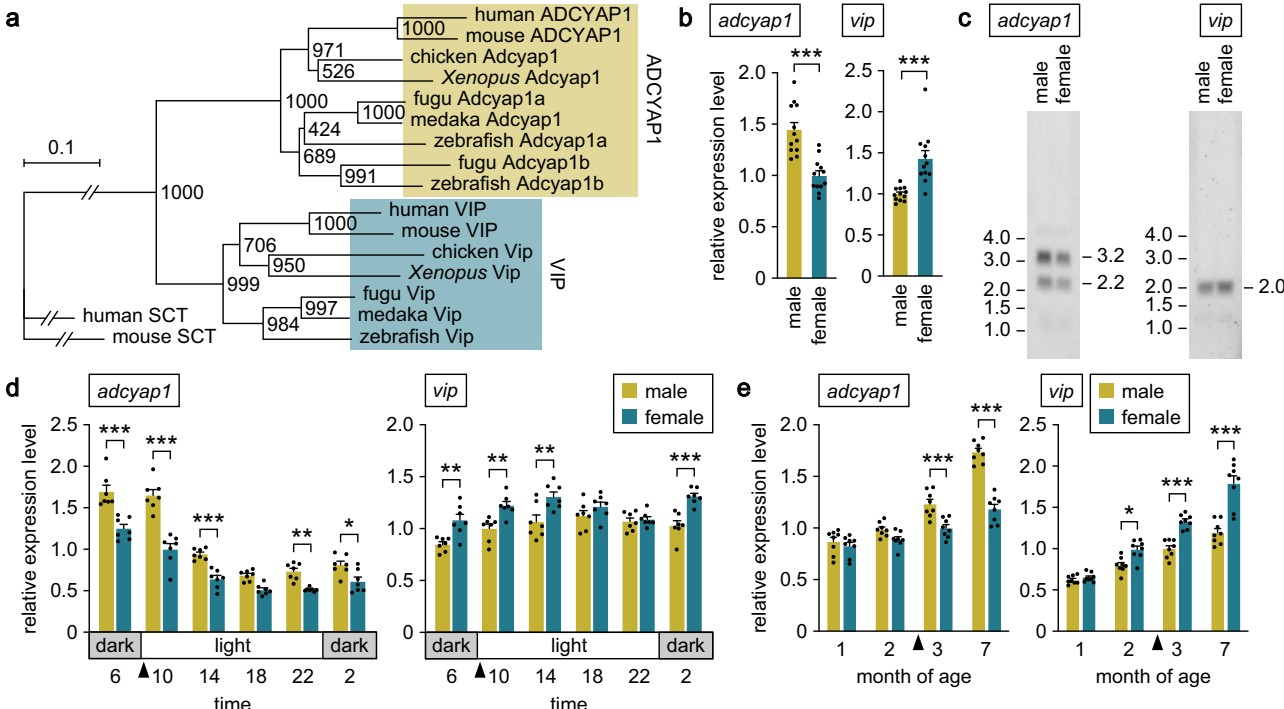

**Fig. 1 adcyap1 and vip exhibit opposite patterns of sex-biased expression in the medaka brain. a** Phylogenetic tree showing the relationship of medaka Adcyap1 and Vip to those of other species. The number at each node indicates bootstrap values for 1000 replicates. Scale bar represents 0.1 substitutions per site. SCT secretin. For species names and GenBank accession numbers, see Supplementary Table 2. **b** Confirmation/identification of sex-biased expression of *adcyap1* and *vip* in the medaka brain by real-time PCR ($n = 12$ per sex). **c** Detection of *adcyap1* and *vip* transcripts in the male and female brain by northern blot analysis. Sizes (in kb) of RNA markers and the detected bands are indicated on the left and right, respectively. **d** Levels of *adcyap1* and *vip* expression in the male and female brain throughout the diurnal/reproductive cycle ($n = 7$ per sex and time). Arrowheads indicate the typical time of spawning. **e** Levels of *adcyap1* and *vip* expression in the male and female brain during growth and sexual maturation ($n = 8$ per sex and age). Arrowheads indicate the typical time of first spawning. Quantitative data were expressed as means with error bars representing standard error of the mean. Statistical differences were assessed by unpaired *t*-test with or without Welch's correction (**b**) and Bonferroni's post-hoc test (**d**, **e**). *$p < 0.05$; **$p < 0.01$; ***$p < 0.001$.

NAT/NVT, NRL, NPT, NRP, PGm and NGp in the hypothalamus; PGZ3 in the optic tectum; TS, MR/IR/SR/IQ, DT/RT, LV and is in the midbrain tegmentum; and gc/ra/RS in the brain stem (Supplementary Fig. 1a–c; abbreviations for medaka brain nuclei are given in Supplementary Table 1). Although there were no sex differences in the spatial patterns, a significantly greater number of *adcyap1*-expressing neurons were observed in males than in females in the Dl/Dm/Dd/Dp, PMp, NC, PGZ3, TS and LV nuclei ($p < 0.0001$, $= 0.0002$, $0.0099$, $< 0.0001$, $< 0.0001$ and $< 0.0001$, respectively) (Fig. 2a–c).

Neurons expressing *vip* were also identified in many brain nuclei, including Dm and Dp in the dorsal telencephalon; Vp in the ventral telencephalon; PMp, PPa and SC in the POA; VM in the thalamus; rHd in the habenula; NPPv, NDIL, NAT, NVT/NRL and NDTL in the hypothalamus; PPd in the pretectum; MR/IR, DT and TS in the midbrain tegmentum; and gc/ra/Ci in the brain stem (Supplementary Fig. 2a–e). Among these nuclei, the PMp and PPa contained a greater number of *vip*-expressing neurons in females than in males ($p < 0.0001$ and $= 0.0074$, respectively) (Fig. 3a–e).

Collectively, these results demonstrate that *adcyap1* and *vip* exhibit opposite patterns of sex-biased expression in a nucleus located in the POA, namely the PMp, which is therefore the major source of the differential expression of these genes.

**Sex-biased expression of *adcyap1* and *vip* in the POA is established by ovarian estrogen.** Next, we explored the mechanisms underlying the sex-biased expression of *adcyap1* and *vip* in the PMp. First, we estimated the magnitude of

chromosomal and gonadal influences on *adcyap1* and *vip* by producing sex-reversed medaka and examining expression of the two genes in the brain. Real-time PCR analysis revealed that sex-reversed XX males exhibited the same high levels of overall *adcyap1* expression as typical XY males, whereas sex-reversed XY females showed significantly lower levels, comparable to those in typical XX females ($p > 0.9999$ and $< 0.0001$ for XX males versus XY males and XX/XY females, respectively; $p = 0.2576$ and $< 0.0001$ for XY females versus XX females and XY males, respectively) (Fig. 4a). A similar pattern of expression was found for *vip*, whose expression levels in sex-reversed XX males and XY females were equivalent to those in typical XY males and XX females, respectively ($p = 0.4258$ and $< 0.0001$ for XX males versus XY males and XX/XY females, respectively; $p > 0.9999$ and $= 0.0007$ for XY females versus XX females and XY males, respectively) (Fig. 4a). Consistent with these results, quantitative in situ hybridization analysis showed that the number of *adcyap1*- and *vip*-expressing neurons in the PMp of sex-reversed XX males and XY females was roughly equivalent to that in typical XY males and XX females, respectively. XX males had significantly more *adcyap1*-expressing neurons and fewer *vip*-expressing neurons in the PMp as compared with XY females ($p < 0.0001$ and $= 0.0003$, respectively) (Fig. 4b, c). These results indicate that the pattern of *adcyap1* and *vip* expression in the PMp is independent of sex chromosome complement but linked to gonadal phenotype.

This finding, together with the increasing sex bias in *adcyap1* and *vip* expression with sexual maturity, led us to hypothesize that sex steroid hormones secreted by the gonads in adulthood

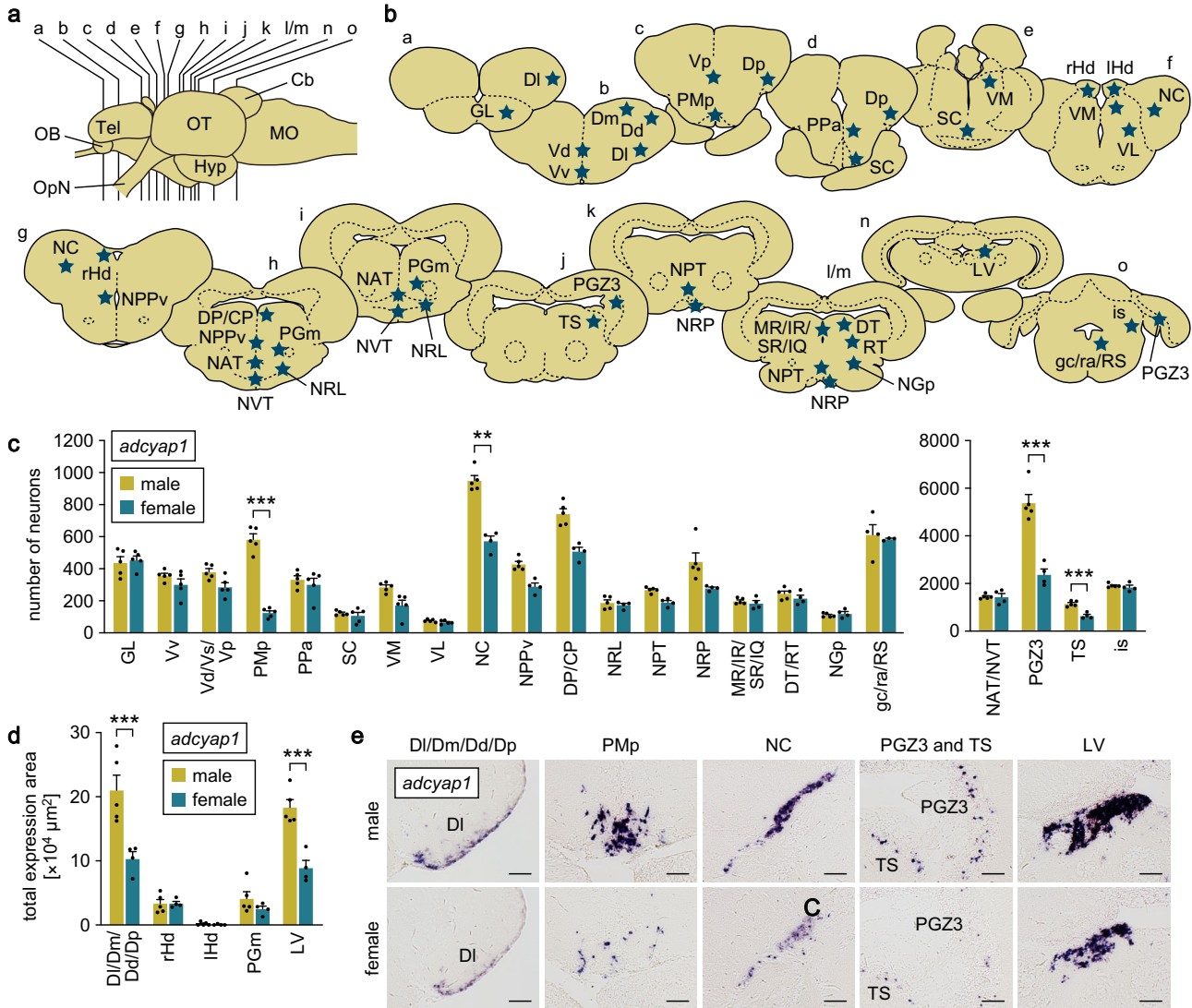

**Fig. 2 The POA is the major source of the sex-biased expression of *adcyap1* in the brain. a** Lateral view (anterior to the left) of the medaka brain showing the approximate levels of sections in panel **b**. **b** Line drawings of coronal brain sections showing the location of nuclei containing *adcyap1*-expressing neurons (stars). **c** Number of *adcyap1*-expressing neurons in each brain nucleus of males ($n = 5$, except for gc/ra/RS, where $n = 4$) and females ($n = 5$ for nuclei from GL to VL; $n = 4$ for nuclei from NC to is, except for gc/ra/RS, where $n = 3$). The data are split into two graphs for visual clarity. **d** Total area of *adcyap1* expression signals in Dl/Dm/Dd/Dp, rHd, lHd, PGm, and LV nuclei, where *adcyap1*-expressing neurons were tightly clustered, and their exact number could not be determined ($n = 5$ and 4 for males and females, respectively). **e** Representative micrographs showing *adcyap1* expression in nuclei where significant differences between the sexes were detected. Scale bars represent 50 μm. Quantitative data were expressed as means with error bars representing standard error of the mean. Statistical differences were assessed by unpaired *t*-test with Bonferroni-Dunn correction (**c**, **d**). **$p < 0.01$; ***$p < 0.001$. For abbreviations of brain nuclei, see Supplementary Table 1.

have a major impact on the pattern of *adcyap1* and *vip* expression in the PMp. To test this idea, we analyzed their expression pattern in fish that were gonadectomized as adults and treated with 11-ketotestosterone (KT; the primary, nonaromatizable androgen in teleosts) or estradiol-17β (E2; the major estrogen in vertebrates including teleosts) by quantitative in situ hybridization. In the male PMp, castration did not cause appreciable changes in numbers of *adcyap1*- or *vip*-expressing neurons, but subsequent treatment with E2 significantly reduced *adcyap1*-expressing neurons ($p = 0.0005$) and increased *vip*-expressing neurons ($p = 0.0006$) (Fig. 4d, e), while KT treatment did not affect either of these neurons. In the female PMp, ovariectomy caused a significant increase in the number of *adcyap1*-expressing neurons ($p = 0.0321$), which was completely abolished by E2 treatment ($p < 0.0001$), whereas KT had no such effect ($p > 0.9999$) (Fig. 4f, g). In contrast, ovariectomy markedly reduced the number of the

*vip*-expressing neurons ($p = 0.0004$), which was restored by E2 treatment ($p = 0.0003$) but not KT treatment ($p = 0.5711$) (Fig. 4f, g).

Taken together, these results suggest that high circulating levels of estrogen derived from the ovary in adulthood attenuate *adcyap1* expression and, conversely, stimulate *vip* expression in the PMp of females, thereby serving to establish and maintain the opposite pattern of sex-biased expression of these genes.

**adcyap1 and vip are expressed in the same population of POA neurons, which are direct targets of estrogen and project to the pituitary.** To test the possibility that estrogen directly acts on *adcyap1*- and *vip*-expressing neurons in the PMp, we determined whether these neurons co-express estrogen receptor (ER). Teleosts, including medaka, have three subtypes of ER, designated

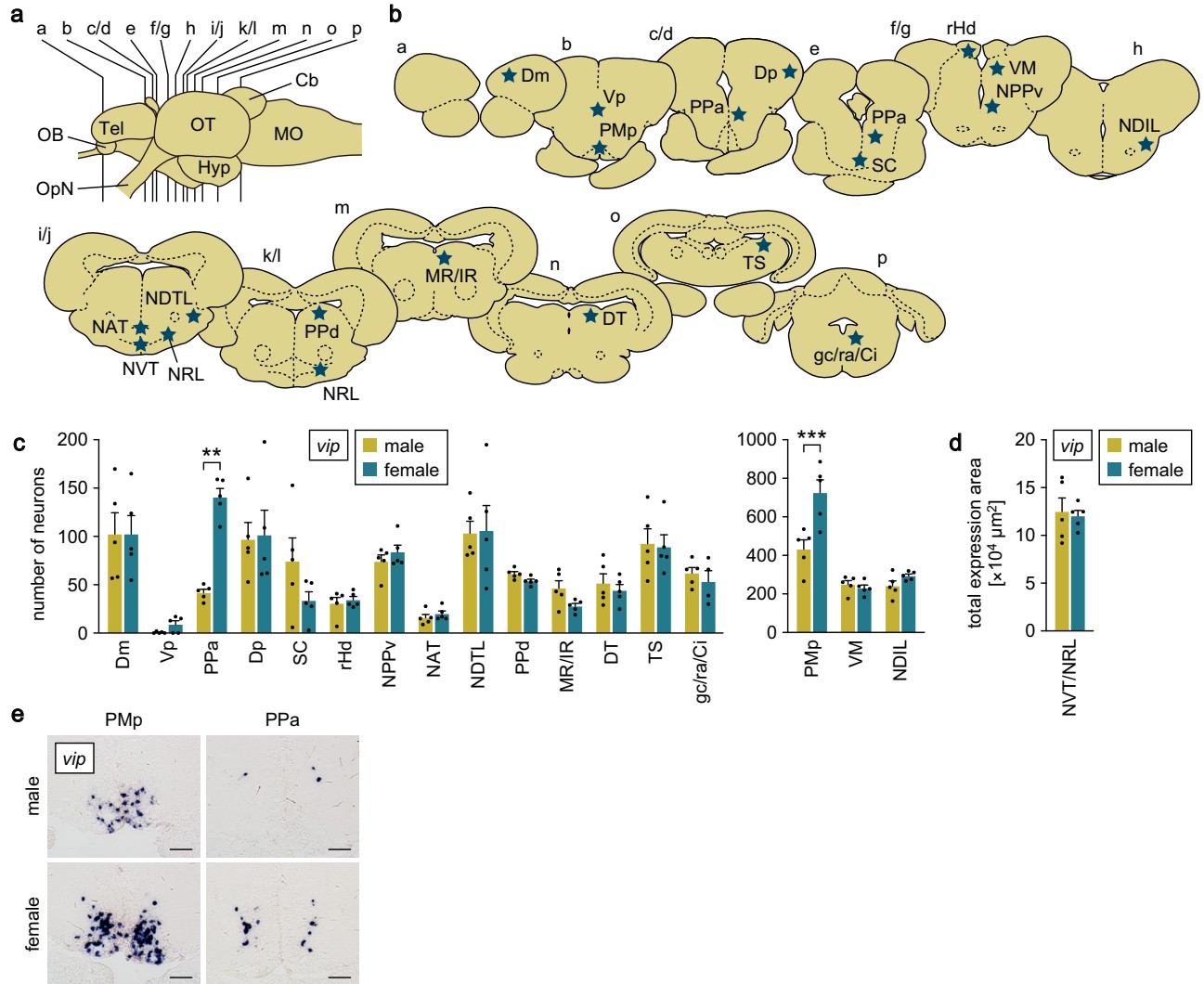

**Fig. 3 The POA is the major source of the sex-biased expression of *vip* in the brain. a** Lateral view (anterior to the left) of the medaka brain showing the approximate levels of sections in panel **b**. **b** Line drawings of coronal brain sections showing the location of nuclei containing *vip*-expressing neurons (stars). **c** Number of *vip*-expressing neurons in each brain nucleus of males and females (n = 5 per sex, except gc/ra/Ci in females, where n = 4). The data are split into two graphs for clarity. **d** Total area of *vip* expression signals in the NVT/NRL nucleus, where *vip*-expressing neurons were tightly clustered, and their exact number could not be determined (n = 5 per sex). **e** Representative micrographs showing *vip* expression in nuclei where significant differences between the sexes were detected. Scale bars represent 50 μm. Quantitative data were expressed as means with error bars representing standard error of the mean. Statistical differences were assessed by unpaired *t*-test with Bonferroni-Dunn correction (**c**, **d**). **\*\*p < 0.01; \*\*\*p < 0.001. For abbreviations of brain nuclei, see Supplementary Table 1.

Esr1, Esr2a and Esr2b[19]. Double in situ hybridization to simultaneously detect the expression of *adcyap1* and each ER subtype revealed that *esr1*, *esr2a* and *esr2b* were all expressed in some, but certainly not all, of the *adcyap1*-expressing neurons in both sexes (Fig. 5a, b). Similarly, the expression of all three ER genes was detected in a proportion of the *vip*-expressing neurons of both sexes (Fig. 5a, c). In both *adcyap1*- and *vip*-expressing neurons, the percentage of neurons co-expressing *esr1* and *esr2b* was significantly higher compared with that of neurons co-expressing *esr2a* in both sexes (p = 0.0071 and 0.0003 for *esr1* and *esr2b*, respectively, in male *adcyap1* neurons; p = 0.0002 and < 0.0001 for *esr1* and *esr2b*, respectively, in female *adcyap1* neurons; p = 0.0016 and 0.0004 for *esr1* and *esr2b*, respectively, in male *vip* neurons; p < 0.0001 for both *esr1* and *esr2b* in female *vip* neurons) (*adcyap1* neurons: main effect of sex, p = 0.0012; main effect of ER subtype, p < 0.0001; interaction between sex and ER subtype, p = 0.1477. *vip* neurons: main effect of sex, p = 0.0248; main effect of ER subtype, p < 0.0001; interaction between sex and ER

subtype, p = 0.0638) (Fig. 5a). Hence, estrogen can potentially act directly, at least partially, on *adcyap1*- and *vip*-expressing neurons and the action of estrogen may be mediated predominantly by *esr1* and *esr2b*.

Next, we examined the axonal projections of *adcyap1*- and *vip*-expressing neurons in the PMp to determine the site of action of Pacap and Vip produced therein. Both types of neurons were mainly located in the anteroventral subregion of the PMp, which also contains neurons that express the neuropeptides isotocin and galanin and project to the anterior pituitary[16, 20] (note that the anterior pituitary of teleosts does not have a median eminence/hypophyseal portal system and receives direct innervation from preoptic/hypothalamic neurons[21, 22]). This observation, together with the established neuroendocrine functions of PACAP and VIP[23–25], led us to assume that the *adcyap1*- and *vip*-expressing neurons also project to the anterior pituitary. This assumption was confirmed by the observation that a substantial proportion of both *adcyap1*- and *vip*-expressing PMp neurons were

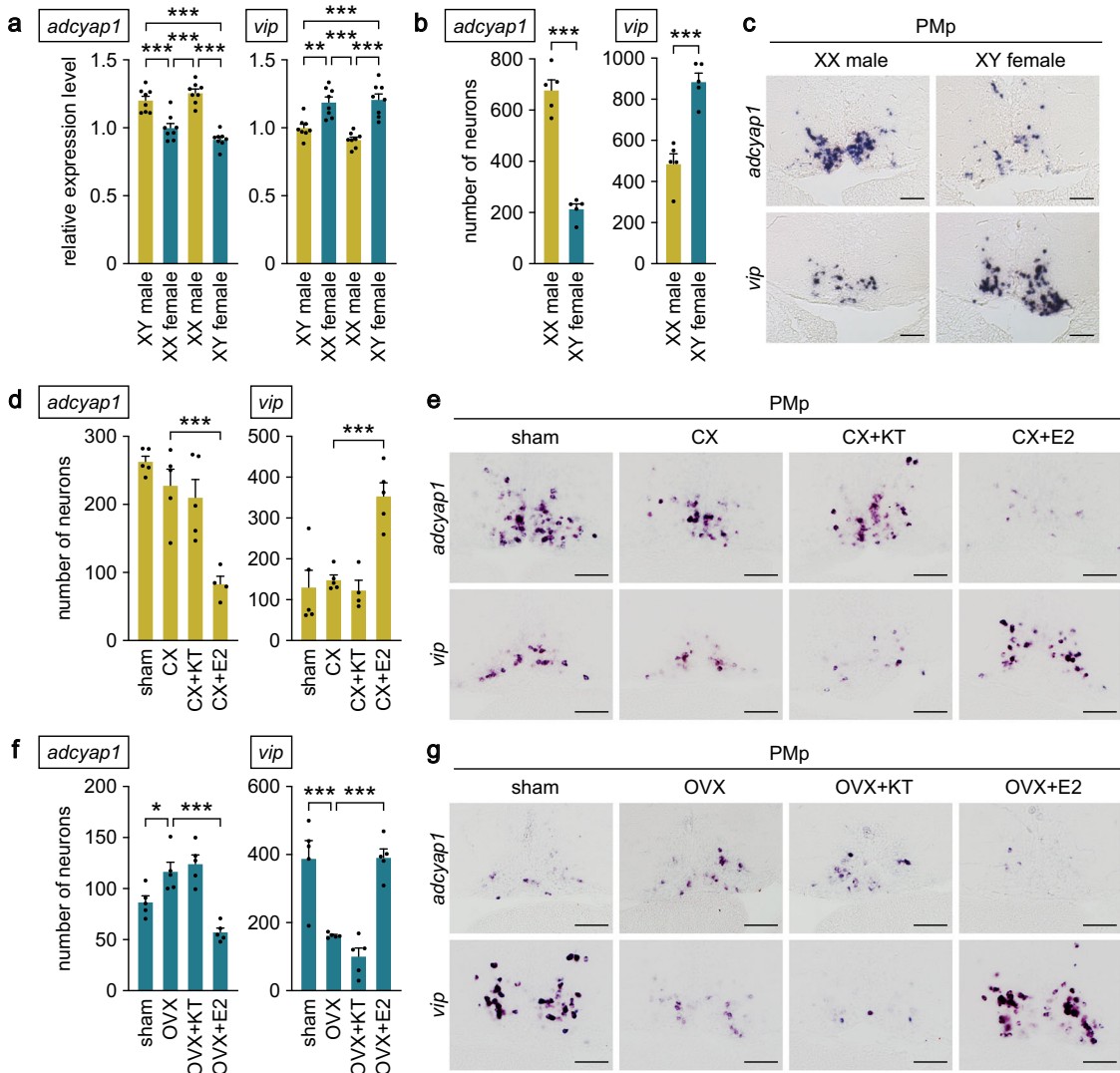

**Fig. 4 Sex-biased expression of *adcyap1* and *vip* in the POA is established by ovarian estrogen. a** Levels of *adcyap1* and *vip* expression in the brain of sex-reversed XX males and XY females versus typical XY males and XX females ($n = 8$ per group). **b** Number of *adcyap1*- and *vip*-expressing neurons in the PMp of sex-reversed XX males and XY females ($n = 5$ per group). **c** Representative micrographs showing *adcyap1* and *vip* expression in the PMp of sex-reversed XX males and XY females. Scale bars represent 50 µm. **d** Number of *adcyap1*- and *vip*-expressing neurons in the PMp of sham-operated (sham) males and castrated males exposed to vehicle alone (CX), 11-ketotestosterone (CX + KT), or estradiol-17β (CX + E2) ($n = 5$ per group, except CX + E2 for *adcyap1* and CX + KT for *vip*, where $n = 4$). **e** Representative micrographs showing *adcyap1* and *vip* expression in the PMp of sham, CX, CX + KT, and CX + E2 males. Scale bars represent 50 µm. **f** Number of *adcyap1*- and *vip*-expressing neurons in the PMp of sham females and ovariectomized females exposed to vehicle alone (OVX), KT (OVX + KT), or E2 (OVX + E2) ($n = 5$ per group). **g** Representative micrographs showing *adcyap1* and *vip* expression in the PMp of sham, OVX, OVX + KT, and OVX + E2 females. Scale bars represent 50 µm. Quantitative data were expressed as means with error bars representing standard error of the mean. Statistical differences were assessed by Bonferroni's post-hoc test (**a**, **d**, **f**) and unpaired *t*-test (**b**). *$p < 0.05$; **$p < 0.01$; ***$p < 0.001$.

retrogradely labeled after the neuronal tracer Neurobiotin was injected into the anterior pituitary (Fig. 6a). In addition, the direct innervation of the anterior pituitary by Pacap- and Vip-containing axons was confirmed by immunohistochemistry, which revealed a dense plexus of axons and varicosities that were intensely immunoreactive for Pacap and Vip in the anterior pituitary (Fig. 6b). There were no noticeable differences in the distribution of axons between the sexes.

In light of these shared characteristics, we wanted to establish whether the *adcyap1*- and *vip*-expressing neurons represent distinct or overlapping populations of neurons in the PMp. Double in situ hybridization for *adcyap1* and *vip* revealed that these neurons were intermingled in the same subregion all along the anterior–posterior extent of the PMp and a proportion of

these neurons expressed both *adcyap1* and *vip* (Fig. 6c). Collectively, these results indicate that Pacap and Vip are expressed in the same neuronal population within the PMp and serve hypophysiotropic functions.

**Multiple Pacap/Vip receptors, including a male-predominant subtype, are expressed in the anterior pituitary.** The hypophysiotropic functions of Pacap and Vip in medaka were further delineated by identifying and characterizing their receptors and by assessing receptor expression in the anterior pituitary. Teleosts have two paralogs each of ADCYAP1R1, VIPR1 and VIPR2 and hence have six Pacap/Vip receptors in total[26, 27]. BLAST searches of the GenBank database identified the following six distinct

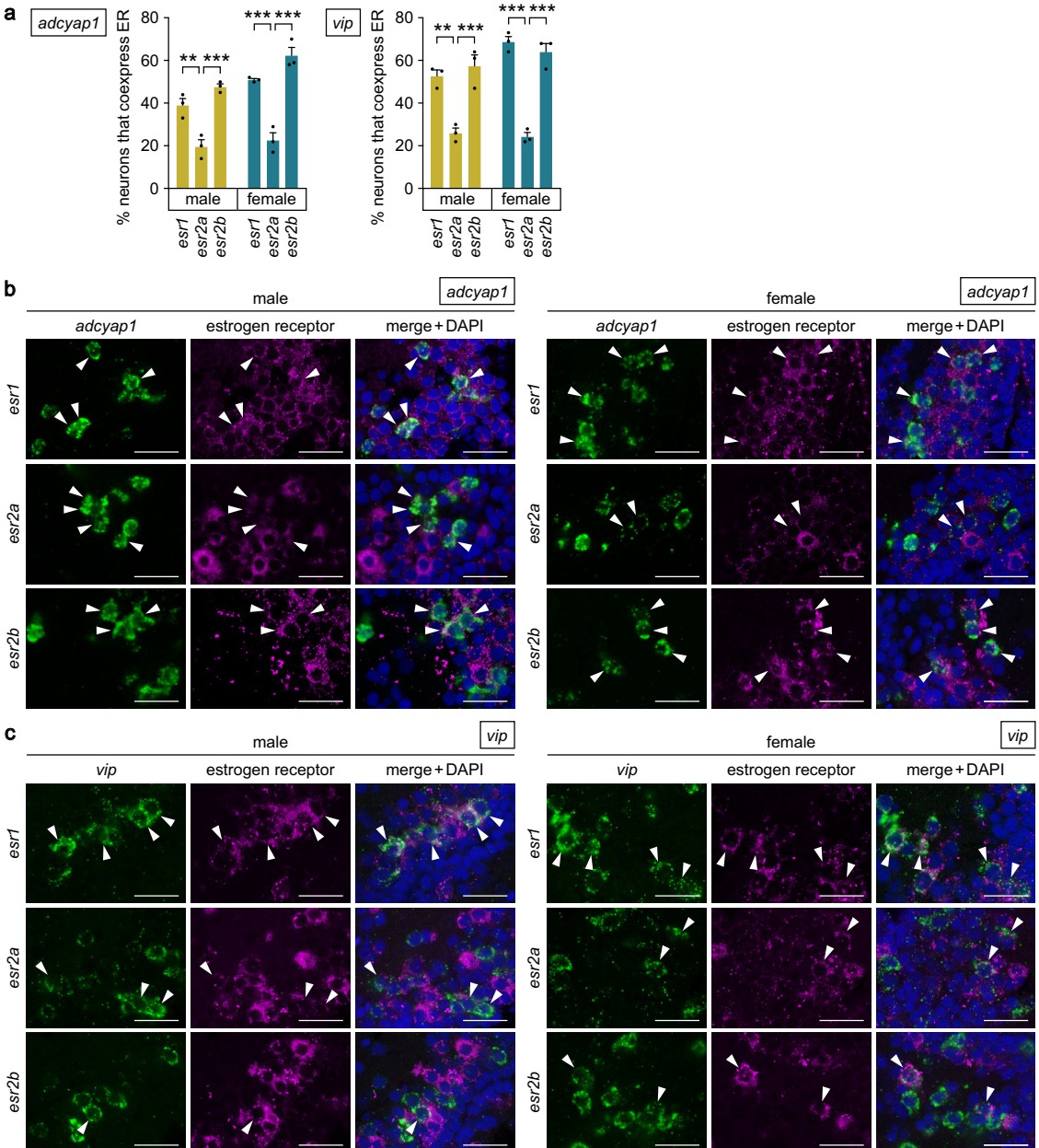

**Fig. 5 *adcyap1*- and *vip*-expressing neurons in the POA are direct targets of estrogen. a** Percentage of *adcyap1*- and *vip*-expressing neurons in the male and female PMp that co-express estrogen receptors (*esr1*, *esr2a*, and *esr2b*) ($n = 3$ fish per group). **b, c** Representative micrographs showing the expression of estrogen receptors in *adcyap1*-expressing (**b**) and *vip*-expressing (**c**) neurons in the male and female PMp. In each row, left and middle panels show images of, respectively, *adcyap1/vip* (green) and estrogen receptor (magenta) expression in the same section; right panel shows the merged image with nuclear counterstaining (blue). Arrowheads indicate representative *adcyap1/vip*-expressing neurons that co-express estrogen receptor. Scale bars represent 20 μm. Quantitative data were expressed as means with error bars representing standard error of the mean. Statistical differences were assessed by Bonferroni's post-hoc test (**a**). **$p < 0.01$; ***$p < 0.001$.

medaka proteins as highly homologous to Pacap/Vip receptors in other teleosts: XP_023820892, XP_004068178, XP_023805948, XP_004079154, XP_023805816 and XP_023821167. Phylogenetic tree analysis further showed that XP_023820892 and XP_004068178 represent Adcyap1r1 (designated Adcyap1r1a and Adcyap1r1b, respectively, according to Cardoso et al.[26], XP_023805948 and XP_004079154 represent Vipr1 (likewise designated Vipr1a and Vipr1b, respectively) and XP_023805816 and XP_023821167 represent Vipr2 (likewise designated Vipr2a and Vipr2b, respectively) (Fig. 7a).

Receptor activation assays revealed that both Pacap and Vip dose-dependently activated all six receptors, resulting in a marked

increase in intracellular cAMP levels, as has been reported for PACAP/VIP receptors of other species[11, 12] (Fig. 7b). However, Pacap exhibited over 100-fold greater potency as compared with Vip in activating Adcyap1r1a and Adcyap1r1b (the half-maximal effective concentration (EC50) of Pacap and Vip was, respectively, 0.2446 and 32.26 nM at Adcyap1r1a and 0.1158 and 17.84 nM at Adcyap1r1b) (Fig. 7b). By contrast, Pacap was approximately 20-fold less potent as compared with Vip in activating Vipr1b (EC50 of 14.85 and 0.7281 nM, respectively) (Fig. 7b). Pacap and Vip had fairly similar potencies at Vipr1a, Vipr2a and Vipr2b, with markedly lower EC50 values at Vipr1a (0.8512 and 0.4269 nM, respectively) relative to Vipr2a (3.556 and

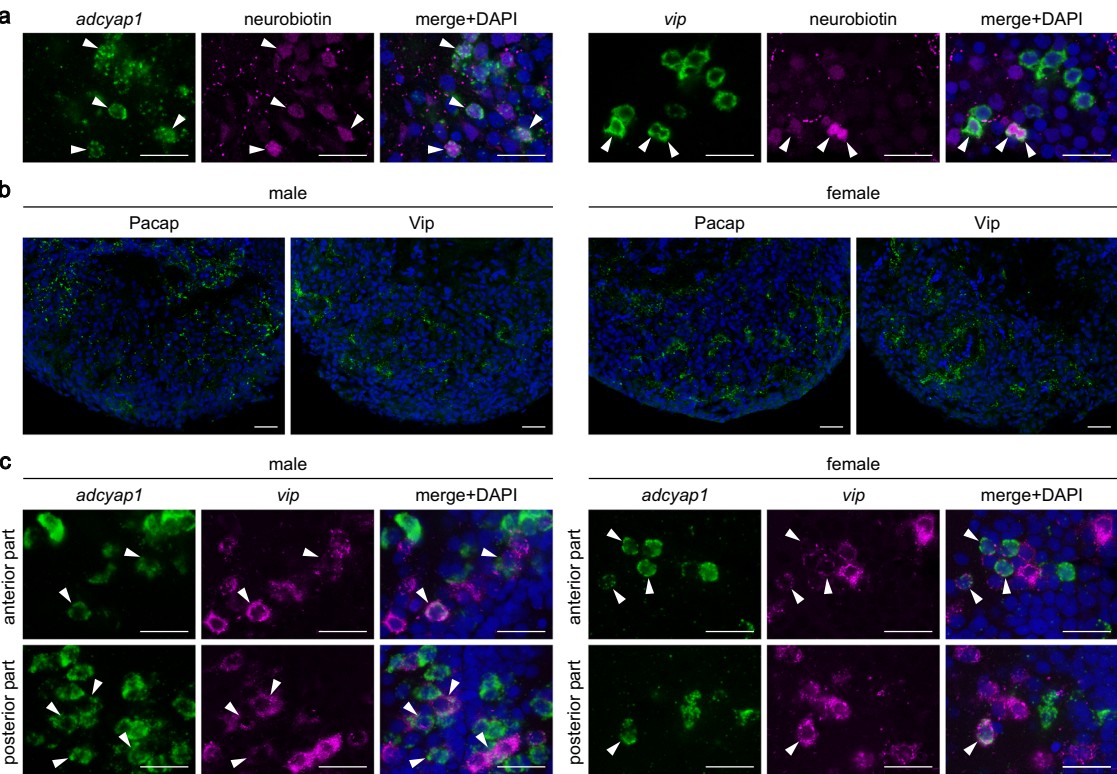

**Fig. 6 *adcyap1* and *vip* are expressed in the same population of POA neurons that project to the pituitary. a** Retrograde labeling of *adcyap1*- and *vip*-expressing neurons in the PMp after injection of the neuronal tracer Neurobiotin into the anterior pituitary. Left and middle panels show images of, respectively, *adcyap1/vip* expression (green) and Neurobiotin labeling (magenta) in the same section; right panel shows the merged image with nuclear counterstaining (blue). Arrowheads indicate representative *adcyap1/vip*-expressing neurons labeled with Neurobiotin. Representative images of *adcyap1*- and *vip*-expressing neurons in a male and female, respectively, are presented. **b** Immunohistochemical detection of Pacap- and Vip-containing axons and varicosities (green) in the male and female anterior pituitary. Blue color indicates nuclear counterstaining. **c** Intermingled and partially overlapping distribution of *adcyap1*- and *vip*-expressing neurons in the male and female PMp. Representative micrographs taken at two different levels along the anterior–posterior axis are shown. Left and middle panels show images of, respectively, *adcyap1* (green) and *vip* (magenta) expression in the same section; right panel shows the merged image with nuclear counterstaining (blue). Arrowheads indicate representative neurons co-expressing *adcyap1* and *vip*. Scale bars represent 20 μm.

2.427 μM, respectively) and Vipr2b (2.775 and 4.311 μM, respectively) (Fig. 7b).

In situ hybridization analysis showed that *adcyap1r1a* was expressed in the rostral pars distalis (RPD) of the pituitary, whereas *adcyap1r1b* and *vipr2b* were expressed in the pars intermedia (PI) (Fig. 7c). These spatial patterns of receptor expression in the pituitary were similar for males and females; however, quantitative analysis revealed a significant male bias in the intensity of *vipr2b* expression ($p = 0.0008$) (Fig. 7d, e). We further investigated the identity of the pituitary cells expressing these receptors. The RPD of teleosts, including medaka, contains prolactin (*prl*)- and pro-opiomelanocortin (*pomc*)-expressing cells, whereas the PI contains somatolactin (*sl*)- and *pomc*-expressing cells[28]. Double in situ hybridization for the receptors and pituitary hormone genes showed that *adcyap1r1a* was expressed in *pomc*-, but not *prl*-expressing, cells in the RPD and *adcyap1r1b* and *vipr2b* were exclusively expressed in *sl*- and *pomc*-expressing cells, respectively, in the PI (Fig. 7f).

In addition to the pituitary, we studied the spatial expression patterns of these receptors in the brain. Expression of *adcyap1r1a* was observed in Dm/Dl/Dp in the dorsal telencephalon; Vv and Vd/Vs/Vp in the ventral telencephalon; PPa and PPp in the POA; VM, VL and DP/CP in the thalamus; NAT/NVT, NRL, NDTL, NDIL, NPT and PGm/NGp in the hypothalamus; is in the midbrain tegmentum; and gc, RI, NIX and NFS/LX in the brain stem (Supplementary Fig. 3a, b). Expression of *adcyap1r1b* was

detected in GL in the olfactory bulb; Dm/Dd/Dl/Dp in the dorsal telencephalon; Vv and Vd/Vs/Vp in the ventral telencephalon; PPa and PPp in the POA; Flt, VM and DP/CP in the thalamus; NC and PPd in the pretectum; NAT/NVT, NRL, NDTL, NDIL, NPT, PGm and NGp in the hypothalamus; PGZ3 in the optic tectum; TS in the midbrain tegmentum; CbSg and CbSm in the cerebellum; and gc, RI and NFS/LX in the brain stem (Supplementary Fig. 3c, d). There were no notable sex differences in *adcyap1r1a* or *adcyap1r1b* expression in the brain. Expression of *vipr1a* was observed in Vv and Vs/Vp in the ventral telencephalon; PPa, PMm and PPp in the POA; VM and DP in the thalamus; lHd in the habenula; NAT/NVT, NRL and NPT in the hypothalamus; and gc, ra and RI in the brain stem (Supplementary Fig. 4a, b). At first glance, there seemed to be female-biased sex differences in *vipr1a* expression in Vv, Vs/Vp, PMm and NPT; therefore, we quantified *vipr1a* expression in these nuclei, which demonstrated significant female bias in Vv and Vs/Vp ($p = 0.0003$ for both nuclei) (Supplementary Fig. 4c, d). Expression of *vipr1b* was hardly detected in the brain, with only weak signals in NAT/NVT in the hypothalamus (Supplementary Fig. 4e, f). Similarly, *vipr2a* expression was detected only weakly in Dd in the dorsal telencephalon and NC in the pretectum (Supplementary Fig. 5a, b). Expression of *vipr2b* was observed in Dm in the dorsal telencephalon; Vv in the ventral telencephalon; PMp and PPa in the POA; choroid plexus; lHd in the habenula; VM and DP/CP in the thalamus; DOT in the

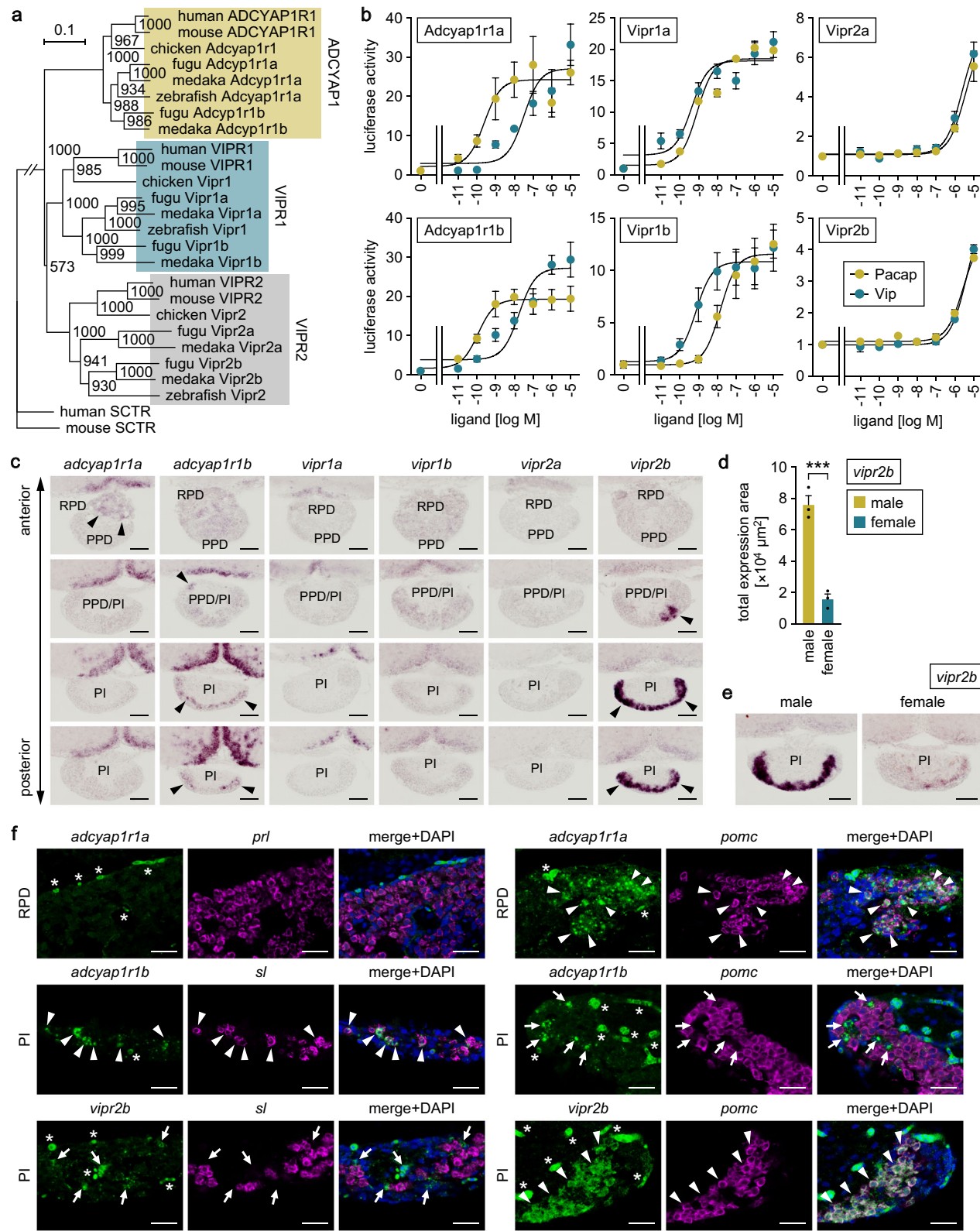

pretectum; NRL, NDTL, PGm and NGp in the hypothalamus; and is, CC and RI in the brain stem (Supplementary Fig. 5c, d).

**Pacap and Vip have pleiotropic effects on the expression of anterior pituitary hormones.** Having established that Pacap and Vip produced in the PMp are transported to the anterior pituitary, where their cognate receptors are expressed, we next

investigated the effects of these peptides on the expression of anterior pituitary hormones by using an ex vivo pituitary organ culture system. Real-time PCR revealed that expression of the follicle-stimulating hormone (Fsh) β subunit (*fshb*) gene was significantly reduced by both Pacap and Vip supplementation in cultured female pituitaries ($p = 0.0176$ and $0.0445$, respectively), but not in male pituitaries (main effect of sex, $p < 0.0001$; main

**Fig. 7 Multiple Pacap/Vip receptors, including a male-predominant subtype, are expressed in the anterior pituitary. a** Phylogenetic tree showing the relationship of medaka Pacap/Vip receptors to those of other species. The number at each node indicates bootstrap values for 1000 replicates. Scale bar represents 0.1 substitutions per site. SCTR, secretin receptor. For species names and GenBank accession numbers, see Supplementary Table 2. **b** Ability of Pacap and Vip to activate medaka Pacap/Vip receptors. Receptor activation was assessed by measuring cAMP-responsive element-driven luciferase activity in cells transfected with the indicated receptors. *x*-axis shows the concentration of Pacap and Vip; *y*-axis shows the fold change in luciferase activity relative to the basal level, which was measured in the absence of Pacap/Vip. **c** Expression of Pacap/Vip receptors in the pituitary. Representative sections of pituitary at different levels along the anterior–posterior axis are shown. Arrowheads indicate expression signals. Scale bars represent 50 μm. **d** Total area of *vipr2b* expression signals in the male and female pituitary (*n* = 3 per sex). **e** Representative micrographs showing male-biased expression of *vipr2b* in the pituitary. Scale bars represent 50 μm. **f** Identity of the pituitary cells expressing each Pacap/Vip receptor. The spatial expression pattern of each receptor was compared with that of each pituitary hormone gene in the RPD and PI. Left and middle panels show images of the expression of the indicated receptor (*adcyap1r1a*, *adcyap1r1b*, or *vipr2b*; green) and pituitary hormone gene (*sl*, *prl*, or *pomc*; magenta) in the same sections; right panels show the merged images with nuclear counterstaining (blue). Arrowheads mark representative cells co-expressing the indicated receptor and pituitary hormone gene, while arrows mark the receptor-expressing cells that do not express the indicated pituitary hormone gene. Asterisks denote nonspecific labeling of blood vessels. Scale bars represent 20 μm. Quantitative data were expressed as means with error bars representing standard error of the mean. Statistical difference was assessed by unpaired *t*-test (**d**). ***p < 0.001. PI pars intermedia, PPD proximal pars distalis, RPD rostral pars distalis.

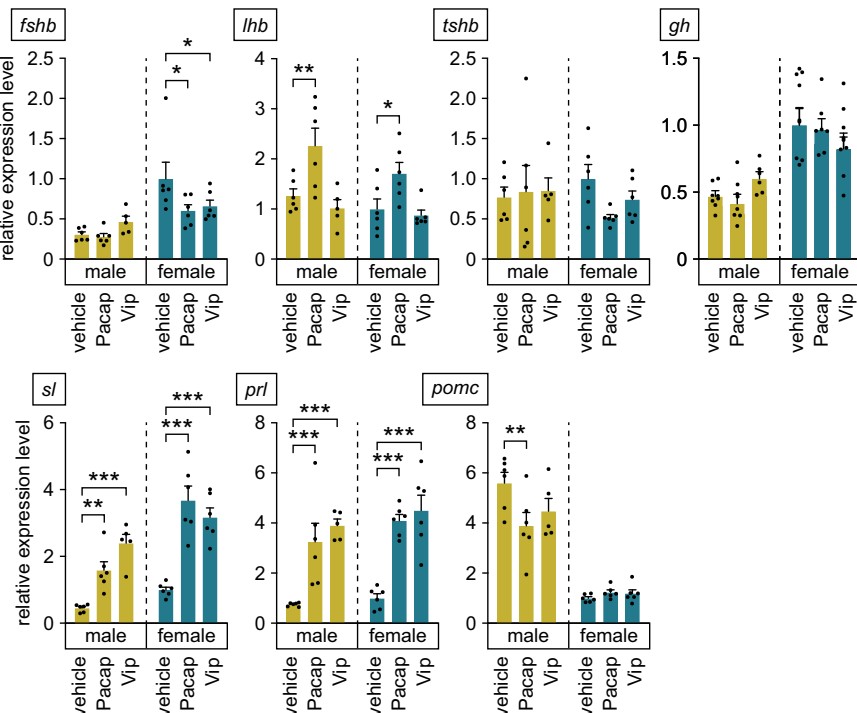

**Fig. 8 Pacap and Vip have pleiotropic effects on the expression of anterior pituitary hormones.** The effects of Pacap and Vip on gene expression levels of follicle-stimulating hormone β subunit (*fshb*), luteinizing hormone β subunit (*lhb*), thyroid-stimulating hormone β subunit (*tshb*), growth hormone (*gh*), somatolactin (*sl*), prolactin (*prl*), and pro-opiomelanocortin (*pomc*) were investigated by using male and female pituitaries cultured ex vivo (*n* = 6 per group, except male pituitary supplemented with Vip, where *n* = 5). Data were expressed as means with error bars representing standard error of the mean. Statistical differences were assessed by Dunnett's post-hoc test. *p < 0.05; **p < 0.01; ***p < 0.001.

effect of treatment, *p* = 0.1252; interaction between sex and treatment, *p* = 0.0557) (Fig. 8). Expression of the luteinizing hormone (Lh) β subunit (*lhb*) gene in both male and female pituitaries was increased by Pacap supplementation (*p* = 0.0045 and 0.0452, respectively), whereas Vip had no significant effect (main effect of sex, *p* = 0.0743; main effect of treatment, *p* < 0.0001; interaction between sex and treatment, *p* = 0.6211) (Fig. 8). Pacap and Vip had no significant effects on expression of the thyroid-stimulating hormone β subunit (*tshb*) or growth hormone (*gh*) gene in either sex (*tshb*: main effect of sex, *p* = 0.6452; main effect of treatment, *p* = 0.5120; interaction between sex and treatment, *p* = 0.3104. *gh*: main effect of sex, *p* < 0.0001; main effect of treatment, *p* = 0.8933; interaction between sex and treatment, *p* = 0.1613) (Fig. 8). The levels of *sl* and *prl* expression were substantially increased by both Pacap and Vip supplementation in both sexes (*sl*: *p* = 0.0079 for Pacap in males and

< 0.0001 for Pacap in females and Vip in both sexes; main effect of sex, *p* < 0.0001; main effect of treatment, *p* < 0.0001; interaction between sex and treatment, *p* = 0.0108. *prl*: *p* = 0.0005 for Pacap in males and < 0.0001 for Pacap in females and Vip in both sexes; main effect of sex, *p* = 0.1199; main effect of treatment, *p* < 0.0001; interaction between sex and treatment, *p* = 0.7814) (Fig. 8). The expression of *pomc* in male pituitaries was reduced by Pacap supplementation (*p* = 0.0025), whereas Vip had no significant effect (*p* = 0.0617). In contrast, neither Pacap nor Vip had no significant effects on *pomc* expression in female pituitaries (main effect of sex, *p* < 0.0001; main effect of treatment, *p* = 0.1068; interaction between sex and treatment, *p* = 0.0239) (Fig. 8).

In summary, both Pacap and Vip have sex-independent stimulatory effects on the expression of *sl* and *prl* and female-specific inhibitory effects on *fshb*. In addition, Pacap exerts sex-

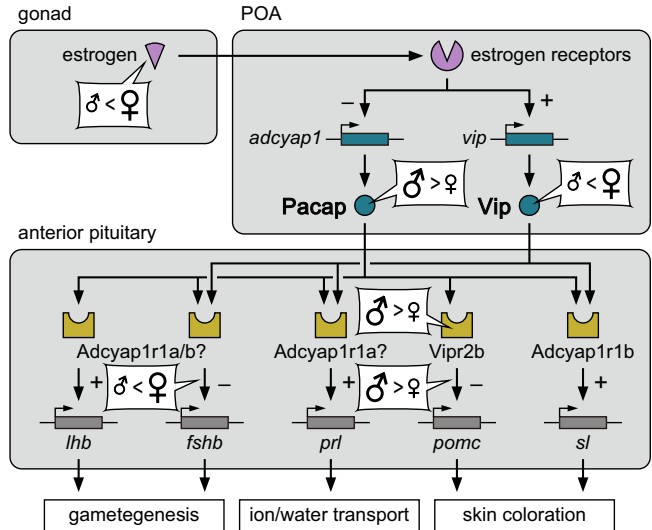

**Fig. 9 Schematic summary illustrating that males and females, respectively, use Pacap and Vip in the POA to control the production of anterior pituitary hormones.** Estrogen secreted by the gonad in adulthood attenuates Pacap expression and, conversely, stimulates Vip expression in the hypophysiotropic population of POA neurons, thereby establishing and maintaining their opposite sexual dimorphism. Both Pacap and Vip have sex-independent stimulatory effects on expression of *prl* and *sl*, and female-specific inhibitory effects on *fshb*. Pacap additionally exerts sex-independent stimulatory effects on *lhb* and male-specific inhibitory effects on *pomc*.

independent stimulatory effects on *lhb* and male-specific inhibitory effects on *pomc*.

## Discussion

A summary of the main findings of this study is presented in Fig. 9. First, we found that in medaka two neuropeptides belonging to the secretin family, Pacap and Vip, have opposite patterns of sex-biased expression in the same population of POA neurons, whereby Pacap is male-biased and Vip is female-biased. Each of these biases is truly sexually dimorphic with little or no overlap in the number of neurons expressing the neuropeptides between males and females. Although various anatomical and molecular sex differences have been identified in the vertebrate brain, particularly in the POA[6, 7], to our knowledge, this is the first study to report opposite patterns of sexual dimorphism in neuropeptide expression in a population of neurons. Given that Pacap and Vip share common receptors, our finding indicates that males and females use different neuropeptides for equivalent signaling functions.

We also found that the sexually dimorphic Pacap- and Vip-expressing neurons in the POA directly project to the anterior pituitary, where functional receptors for these peptides are expressed. This suggests that Pacap and Vip produced therein act as hypophysiotropic factors. In rodents, the hypophysiotropic population of PACAP neurons resides in the paraventricular nucleus (PVN) of the hypothalamus rather than in the POA[24]. Therefore, Pacap- and Vip-expressing neurons in the medaka POA may be functionally homologous to those in the PVN of rodents. In rat and cat, however, PACAP and VIP are not co-expressed in the same neurons in any brain region, including the PVN[29]. More notably, the hypophysiotropic effects of neuronally produced VIP are assumed to be indirect and mediated by other hypothalamic factors in mammals[30]. These facts suggest that mammals may not have the exact equivalent of Pacap- and Vip-expressing neurons that are present in the medaka POA. Further comparative studies across multiple species are needed to

determine the extent to which these neurons are conserved among vertebrates.

Importantly, our results revealed that the sexually dimorphic expression of both Pacap and Vip is caused by adult secretion of estrogen; that is, estrogen secreted by the ovary in adulthood attenuates *adcyap1* expression and, conversely, stimulates *vip* expression in the female POA. Consistent with this observation, we also found that sex differences in *adcyap1* and *vip* expression decrease after the mid-light phase, when blood estrogen levels in females are at their lowest[31, 32]. Together with the above findings, this observation reveals that, owing to their different estrogen milieu, males and females employ different preoptic neuropeptides that act through common receptors but respond to estrogen in opposite ways to regulate anterior pituitary function. Our results also showed that the Pacap- and Vip-expressing neurons in the POA express all three ER subtypes, with Esr1 and Esr2b being predominant, indicating that estrogen acts directly on these neurons to affect the expression of these peptides. In rat and quail (*Coturnix coturnix*), neural expression of *Vip* is female-biased and estrogen-dependent[33–35], as found in medaka. This suggests that the stimulatory role of estrogen on VIP expression is evolutionarily conserved in vertebrates. By contrast, there is evidence that in rodents, unlike in medaka, estrogen does not affect *Adcyap1* expression in the POA but positively regulates it in other brain regions, including the ventromedial hypothalamus and the bed nucleus of the stria terminalis[36, 37]. In addition, PACAP concentrations in hypophyseal portal blood are two-fold higher in female rats than in male rats[38], suggesting female-biased expression of PACAP in hypophysiotropic PVN neurons. Thus, estrogenic regulation of PACAP expression may have changed during vertebrate evolution.

Using an ex vivo pituitary organ culture system, we demonstrated that Pacap and Vip stimulate the expression of *sl* and *prl* in the pituitaries of both sexes and suppress the expression of *fshb* only in female pituitaries. Pacap was additionally found to stimulate *lhb* in both sexes and suppress *pomc* only in males. These results are mostly in line with studies in other species showing that PACAP and VIP have potent modulatory effects on anterior pituitary hormones, although the specific patterns of effects vary across studies[11, 23–25, 39, 40]. Combined with the above findings, these results suggest that these pituitary hormone genes are differentially controlled in males and females and that Pacap and Vip, respectively, are primarily responsible for this hormonal regulation.

The stimulatory effects of Pacap and Vip on *sl* and *prl* were comparable between male and female pituitaries and therefore are probably mediated by *adcyap1r1a* and/or *adcyap1r1b*, which are expressed equivalently in male and female pituitaries. Combining this with our additional finding that *sl*-expressing cells express *adcyap1r1b*, Pacap and Vip may act directly on these cells to stimulate *sl* expression via Adcyap1r1b. Consistent with this notion, in grass carp (*Ctenopharyngodon idellus*) and goldfish (*Carassius auratus*), Pacap has been shown to act directly on Sl-producing cells and to stimulate the synthesis and release of Sl via Adcyap1r1[41–43]. In contrast, since no expression of Pacap/Vip receptors was detected in *prl*-expressing cells, it seems unlikely that Pacap and Vip have direct effects on *prl* expression. Considering that *pomc*-expressing cells in the RPD adjacent to *prl* cells express *adcyap1r1a*, Pacap and Vip may stimulate *prl* expression through their action on these *pomc* cells. Sl is a fish-specific member of the Gh/Prl family of pituitary hormones, which has been implicated in skin coloration[44]. Across vertebrates, Prl is a key hormonal regulator of ion and water transport in various tissues, including those responsible for whole-organism ion homeostasis[45, 46]. These functions of Sl and Prl are equally important for both sexes. It is possible that regulation of *sl* and *prl*

expression is mediated by estrogen-repressed and estrogen-stimulated neuropeptides in males and females, respectively, in order to ensure the function of these pituitary hormones in both sexes regardless of the estrogen milieu. Pacap and Vip were equally effective in stimulating *sl* and *prl* expression, probably because both neuropeptides were administered at a relatively high concentration. However, the potency of Pacap was more than 100-fold higher than that of Vip in activating both Adcyap1r1a and Adcyap1r1b in medaka, as has been reported in other species[11, 12] and thus is presumably more effective than Vip at lower concentrations. Additional testing with lower concentrations of Pacap and Vip will be necessary to verify this presumption. Given that *sl* is expressed in a female-biased fashion (this and previous studies[47]), Pacap may, by acting predominantly in males, serve to reduce this sex difference, which would be otherwise substantial.

In contrast, the inhibitory effects of Pacap on *pomc* were evident only in male pituitaries and thus seem to be mediated by *vipr2b*, which is expressed more abundantly in male than in female pituitaries. Given the expression of *vipr2b* by *pomc*-expressing cells, Pacap may act directly on these cells to repress the expression of *pomc*. Although activation of Vipr2b requires rather high concentrations of Pacap, the immunohistochemical detection of a dense plexus of Pacap-containing axons and varicosities in the pituitary suggests that Pacap can act locally at high concentrations in the pituitary to activate Vipr2b. Pomc is post-translationally processed in the PI to yield α-melanocyte-stimulating hormone, whose primary function in teleosts is to regulate skin coloration[48]. Consistent with the fact that male medaka generally have darker and more vivid skin color than females, the basal expression of *pomc* is male-biased (this and previous studies[47]). Similar to *sl*, the male-dominant action of Pacap may serve to reduce the sex difference in *pomc* expression, which would otherwise be even larger. In other words, Pacap may prevent extreme skin darkening in males by suppressing *pomc*. Medaka prefer mates with skin color similar to their own and avoid mates with skin color that differ greatly from their own, suggesting that premating reproductive isolation is occurring between strains of medaka with different skin colors[49]. The differential use of Pacap and Vip may contribute to this process by reducing sex differences in skin color through the regulation of *sl* and *pomc*.

Contrary to *pomc*, *fshb* expression was female-biased and repressed by Pacap and Vip in a female-specific manner. The inhibitory effect of PACAP on *Fshb* has also been reported in rat[24, 25], suggesting the evolutionary conservation of this effect. Studies on dispersed rat pituitary cells indicate that PACAP acts directly on gonadotrophs to inhibit *Fshb* expression and that this effect is mediated by increased expression of follistatin[24, 25]. Possibly, Pacap may have a similar mechanism of action in medaka. In medaka, however, the inhibitory effect on *fshb* is assumed to be primarily driven by Vip, which acts predominantly in females. Similar to the role of Pacap in *pomc* expression, Vip may contribute to reducing the sex difference in *fshb* expression. Our data also showed that Pacap, but not Vip, stimulated the expression of *lhb*, which encodes another gonadotropin β-subunit. Why only Pacap showed a stimulatory effect is uncertain, but it is possible that Adcyap1r1a or Adcyap1r1b, both of which are activated more potently by Pacap than by Vip, is expressed at low levels in *lhb*-expressing cells (although we were unable to detect this expression) and mediates the effect of Pacap on *lhb* (note that in teleosts, unlike in other vertebrates, *fshb* and *lhb* are expressed in different cell populations[21, 22]). This possibility is supported by the finding that Pacap directly stimulates the release of luteinizing hormone via Adcyap1r1 in both goldfish and rat[50]. The expression levels of both *fshb* and *lhb* have been

shown to fluctuate with the diurnal/reproductive cycle of female medaka[51]. These fluctuations are attributed to the cyclic nature of the action of gonadotropin-releasing hormone 1[51], but it may be worthwhile in future studies to assess whether Pacap and Vip are also involved in these fluctuations.

In summary, the present study has revealed that males and females use different preoptic neuropeptides to regulate anterior pituitary hormones as a consequence of their different estrogen milieu. It is becoming increasingly accepted that not all sex differences in the brain cause sex differences in physiological and behavioral traits and that some of them serve to prevent sex differences in these traits[9, 52, 53]. The differential expression of neuropeptides for brain–pituitary signaling identified in the present study is considered a typical example of such sex differences and most probably serves to prevent undesirable sex differences in gametogenesis, ion and water transport and skin coloration. The differential use of neuropeptides that are oppositely regulated by estrogen may represent a regulatory mechanism that has evolved to ensure these pituitary functions regardless of the estrogen milieu. It would be worth exploring in future studies whether, and to what extent, the present findings in medaka are applicable to other species.

## Methods

**Animals**. The care and use of animals were done in accordance with the guidelines of the Institutional Animal Care and Use Committee of the University of Tokyo. Medaka of the d-rR strain were bred and maintained at 28 °C with a 14 h light/10 h dark cycle. They were fed 3–4 times per day with live brine shrimp and commercial pellet food (Otohime; Marubeni Nissin Feed, Tokyo, Japan). Spawning adult fish (aged 2–4 months) were used in all analyses except for the determination of *adcyap1/vip* expression levels during growth and sexual maturation, where fish aged 1, 2, 3 and 7 months were used. Fish were sampled at 1–3 h after onset of the light period in all analyses except for the determination of *adcyap1/vip* expression levels throughout the diurnal/reproductive cycle, where they were sampled six times a day at 4 h intervals.

**Production of sex-reversed medaka**. Fertilized eggs were treated with 0.2 ng ml⁻¹ methyltestosterone (Fujifilm Wako Pure Chemical Corporation, Osaka, Japan) at high temperature (32 °C) until hatching for the production of XX males, or treated with 200 ng ml⁻¹ E2 (Fujifilm Wako Pure Chemical Corporation) at normal temperature (28 °C) until hatching for the production of XY females.

**Gonadectomy and drug treatment**. For both males and females, the gonad was removed under 0.02% tricaine methane sulfonate anesthesia (Sigma–Aldrich, St. Louis, MO, USA) through a small incision in the ventrolateral abdominal wall. Immediately after removal of the gonad, the incision was sutured with nylon thread. Sham-operated fish underwent the same surgical treatment as gonadectomized fish except for gonad removal. After a recovery period of 3 days in saline (0.9% (w v⁻¹) NaCl), gonadectomized fish were immersed in water containing 100 ng ml⁻¹ of KT (Cosmo Bio, Tokyo, Japan) or E2 (Fujifilm Wako Pure Chemical Corporation), or vehicle (ethanol) alone for 6 days and then sampled. Sham-operated fish were treated with vehicle alone and used as controls. The sex steroid concentration used was based on previously reported serum steroid levels in medaka[54].

**Suppression subtractive hybridization**. Suppression subtractive hybridization was performed previously and reported in Okubo et al.[18]. In brief, poly(A)⁺ RNA isolated from male and female brains was subjected to subtractive hybridization and suppressive PCR by using the PCR-Select cDNA Subtraction Kit (Takara Bio, Shiga, Japan). In total, 3072 clones were randomly picked and sequenced from each of the male- and female-enriched cDNA libraries. After clustering and assembly, 768 clusters of clones were chosen for further screening by dot-blot analysis using the AlkPhos Direct Labeling and Detection System with CDP-Star (Cytiva, Marlborough, MA, USA).

**cDNA cloning and phylogenetic tree analysis**. A full-length cDNA library was constructed from the medaka brain and approximately 32,000 clones were randomly selected and sequenced in the 5′ to 3′ direction as described elsewhere[18]. After assembly and annotation, two clones were identified that had best BLAST hits to *adcyap1* in other species. One of the clones (ID: 44-M02) was fully sequenced. To obtain the full-length cDNA for *vip*, RACE was performed on medaka brain poly(A)⁺ RNA using the Marathon cDNA Amplification Kit (Takara Bio) as described previously[55]. The resulting amplicons were subcloned into the pGEM-T

Easy vector (Promega, Madison, WI, USA) and sequenced. All sequences were determined from analysis of at least three independent amplicons.

The deduced amino acid sequences of Adcyap1/Vip and their receptors in medaka were aligned with those of other species by using ClustalW. The resulting alignments were used to construct bootstrapped (1000 replicates) neighbor-joining trees (http://clustalw.ddbj.nig.ac.jp/index.php). Human and mouse secretin and secretin receptor were used as outgroups to root the trees of ADCYAP1/VIP and their receptors, respectively. The species names and GenBank accession numbers of the sequences used are listed in Supplementary Table 2.

**Real-time PCR on brain samples**. Total RNA was isolated from male and female brains by using the RNeasy Lipid Tissue Mini Kit (Qiagen, Hilgen, Germany) with DNase treatment (Qiagen). cDNA was synthesized by using the Omniscript RT Kit (Qiagen). Real-time PCR was performed on the ABI Prism 7000 Sequence Detection System using Power SYBR Green PCR Master Mix (Thermo Fisher Scientific, Waltham, MA, USA). The cycle conditions were: 95 °C for 10 min followed by 40 cycles of 95 °C for 15 sec and 60 °C for 1 min. For every reaction, melting curve analysis was conducted to ensure that a single amplicon was produced in each sample. The β-actin gene (*actb*; GenBank accession number NM_001104808) was used to normalize the levels of target transcripts in each sample. The suitability of *actb* as a reference gene in the medaka brain has been demonstrated previously[20]. The primers used for real-time PCR are listed in Supplementary Table 3.

**Northern blot analysis**. Northern blot analysis was performed as described previously[16]. In brief, 7 μg of poly(A)$^+$ RNA isolated from male and female brains was separated on a denaturing agarose gel and transferred to a nylon membrane (Roche Diagnostics, Basel, Switzerland). Fragments of 1525 and 725 bp corresponding to, respectively, nucleotides 501–2025 of the medaka *adcyap1* cDNA (deposited in GenBank with accession number LC579549) and nucleotides 84–808 of the medaka *vip* cDNA (deposited in GenBank with accession number LC579550) were PCR-amplified and transcribed in vitro to generate digoxigenin (DIG)-labeled cRNA probes by using DIG RNA Labeling Mix and T7 RNA polymerase (Roche Diagnostics). The resulting probes were hybridized to the membrane in DIG Easy Hyb (Roche Diagnostics) at 65 °C overnight. Hybridization signals were detected with alkaline phosphatase-conjugated anti-DIG antibody and CDP-Star (Roche Diagnostics). Chemiluminescent images were captured by using a LAS-3000 Mini chemiluminescence imaging system (Fujifilm Wako Pure Chemical Corporation).

**Single-label in situ hybridization**. DNA fragments corresponding to nucleotides 2900–4104 (1205 bp) of the *adcyap1r1a* cDNA (GenBank accession number XM_023965124), 1849–3168 (1320 bp) of the *adcyap1r1b* cDNA (XM_011474272), 1545–2875 (1331 bp) of the *vipr1a* cDNA (XM_023950180), 985–2307 (1323 bp) of the *vipr1b* cDNA (XM_004079106), 500–1824 (1325 bp) of the *vipr2a* cDNA (XM_023950048) and 512–1585 (1074 bp) of the *vipr2b* cDNA (XM_023965399) with an extended 3′ end (8007489–8007743 of CP020681) (1329 bp in total) were PCR-amplified and used to generate DIG-labeled cRNA probes as described above.

The procedure for single-label in situ hybridization has been described elsewhere[47]. In brief, the brain and pituitary were fixed in 4% paraformaldehyde (PFA) and embedded in paraffin. Serial coronal sections of 10 μm thickness were cut and hybridized with each of the six DIG-labeled Adcyap1/Vip receptor probes described above and the DIG-labeled *adcyap1* and *vip* probes used for northern blot analysis. Hybridization signals were visualized by using alkaline phosphatase-conjugated anti-DIG antibody and 5-bromo-4-chloro-3-indolyl phosphate/nitro blue tetrazolium (BCIP/NBT) substrate (Roche Diagnostics). Color development was allowed to proceed for more than 9 h or was stopped after 1.5 h to avoid saturation (for quantification of *vip* expression after gonadectomy and sex steroid replacement). All sections in each comparison were processed simultaneously under the same conditions.

To obtain quantitative data, the number of neurons positive for a hybridization signal was counted manually in each brain nucleus. In some brain regions, these neurons were continuously distributed across two or more nuclei such that it was difficult to count the number of the neurons separately in each nucleus. In this case, the total number of the neurons in the nuclei was counted. For the pituitary and nuclei where these neurons were tightly clustered and their exact number could not be determined, sections were photographed and converted to black and white binary images by thresholding using Adobe Photoshop (ver. 22; Adobe Systems, San Jose, CA, USA). The total area of hybridization signal was then calculated by using ImageJ (http://rsbweb.nih.gov/ij/). Brain nuclei were identified by using medaka brain atlases[56,57], supplemented with information from Nissl-stained sections[55].

**Double-label in situ hybridization**. Double-label in situ hybridization was essentially done as described by Kawabata-Sakata et al.[47]. In brief, the brain and pituitary were fixed in 4% PFA and embedded in paraffin (for calculation of the percentage of *adcyap1/vip*-expressing neurons that co-express ER genes and co-expression analysis of Pacap/Vip receptor and pituitary hormone genes) or 5% agarose (Type IX-A; Sigma–Aldrich) supplemented with 20% sucrose for cryoprotection (for co-expression analysis of *adcyap1/vip* and ER genes and that of *adcyap1* and *vip*). Paraffin sections of 10 μm thickness or frozen sections of 20 μm thickness were cut in the coronal plane and hybridized simultaneously with fluorescein- and DIG-labeled cRNA probes. The *adcyap1*, *vip* and Pacap/Vip receptor probes mentioned above and the ER (*esr1*, XM_020714493; *esr2a*, NM_001104702; *esr2b*, NM_001128512) probes described previously[19] were labeled with fluorescein by using Fluorescein RNA Labeling Mix and T7 RNA polymerase (Roche Diagnostics). The above-mentioned *vip* and ER probes and the pituitary hormone (*fshb*, NM_001309017; *lhb*, NM_001137653; *tshb*, XM_004068796; *gh*, XM_004084500; *sl*, NM_001104790; *prl*, XM_004071867; *pomc*, XM_004066456) probes described previously[47] were labeled with DIG as described above. The fluorescein-labeled probe was visualized by using horseradish peroxidase-conjugated anti-fluorescein antibody and the TSA Plus Fluorescein System (PerkinElmer, Waltham, MA, USA); the DIG-labeled probe was visualized by using anti-DIG mouse primary antibody (Abcam, Cambridge, UK) and Alexa Fluor 555-conjugated goat anti-mouse IgG secondary antibody (Thermo Fisher Scientific) (for calculation of the percentage of *adcyap1/vip*-expressing neurons that co-express ER genes and co-expression analysis of Pacap/Vip receptor and pituitary hormone genes) or by using alkaline phosphatase-conjugated anti-DIG antibody and Fast Red (Roche Diagnostics) (for co-expression analysis of *adcyap1/vip* and ER genes and that of *adcyap1* and *vip*). Cell nuclei were counterstained with 4′,6-diamidino-2-phenylindole (DAPI). Fluorescent images were acquired by using a confocal laser scanning microscope (Leica TCS SP8; Leica Microsystems, Wetzlar, Germany). The following excitation and emission wavelengths were used for detection: DAPI, 405 nm and 410–480 nm, respectively; fluorescein, 488 nm and 495–545 nm, respectively; and Fast Red and Alexa Fluor 555, 552 nm and 620–700 nm, respectively.

In the calculation of the percentage of *adcyap1* and *vip* neurons that co-express ER, four or five PMp sections per individual were photographed and examined for ER expression in all *adcyap1* and *vip* neurons in the sections (a total of more than 30 neurons for each individual).

**Retrograde neuronal tracing**. Retrograde axonal tracing of POA neurons projecting to the anterior pituitary was performed essentially as described previously[51]. The brain with the pituitary attached was obtained and the neuronal tracer Neurobiotin (Vector laboratories, Burlingame, CA, USA) was injected into the anterior pituitary. After incubation in fish artificial cerebrospinal fluid for 5 min, the specimen was fixed in 4% PFA and embedded in 5% agarose (Type IX-A; Sigma–Aldrich) supplemented with 20% sucrose. Frozen 20 μm thick coronal POA sections were cut, hybridized with the fluorescein-labeled *adcyap1* or *vip* cRNA probe and visualized as described above. Neurobiotin was visualized by using Alexa Fluor 555-conjugated streptavidin (Thermo Fisher Scientific) and the Vectastain ABC-AP Kit (Vector Laboratories). Fluorescent images were obtained as described above.

**Immunohistochemistry**. The pituitary was fixed in 4% PFA and embedded in 5% agarose (Type IX-A; Sigma–Aldrich) supplemented with 20% sucrose. Frozen 20 μm thick sections were cut in the coronal plane. After blocking with phosphate-buffered saline (PBS) containing 2% normal goat serum, the sections were incubated overnight at 4 °C with anti-PACAP antibody (RRID: AB_519166; Peninsula Laboratories, San Carlos, CA, USA) or anti-VIP antibody (RRID: AB_572270; ImmunoStar, Hudson, WI, USA) diluted at 1:500 or 1:1000, respectively, in PBS containing 2% normal goat serum, 0.1% bovine serum albumin and 0.02% keyhole limpet hemocyanin. The sections were then reacted overnight at 4 °C with Alexa Fluor 488-conjugated goat anti-rabbit IgG (Thermo Fisher Scientific) diluted 1:1000 in PBS. The excitation and emission wavelengths for Alexa Fluor 488 were 488 nm and 495–545 nm, respectively.

The anti-PACAP and VIP antibodies used have been shown to recognize teleost Pacap and Vip, respectively, with high specificity[58–62]. The specificity of these antibodies was further verified by preabsorbing with the synthetic medaka Pacap and Vip polypeptides (see below), which blocked immunodetection.

**Receptor activation assay**. Receptor activation assay was performed essentially as described previously[16]. Medaka Pacap (38 amino acids) and Vip (28 amino acids) polypeptides with amidated C termini were synthesized by Scrum (Tokyo, Japan). cDNA fragments encoding the full-length Adcyap1r1a (GenBank accession number XM_023965124), Adcyap1r1b (XM_011474272), Vipr1a (XM_023950180), Vipr1b (XM_004079106), Vipr2a (XM_023950048) and Vipr2b (XM_023965399) were PCR-amplified and subcloned into the expression vector pcDNA3.1/V5-His-TOPO (Thermo Fisher Scientific). Each of the resulting receptor expression constructs was transiently transfected into COS-7 cells (obtained from and authenticated by Riken BRC Cell Bank), together with the cAMP-responsive luciferase reporter vector pGL4.29 (Promega) and the internal control vector pGL4.74 (Promega) at a ratio of 11:18:1 for 6 h by using Lipofectamine LTX and Plus reagent (Thermo Fisher Scientific). Forty-two hours after transfection, cells were stimulated with Pacap or Vip at doses of 0, $10^{-11}$, $10^{-10}$, $10^{-9}$, $10^{-8}$, $10^{-7}$, $10^{-6}$ and $10^{-5}$ M for 6 h. After cell lysis, luciferase activity was measured by using the Dual-Luciferase Reporter Assay System (Promega). Each assay was performed in triplicate and repeated three times independently.

**Pituitary organ culture and real-time PCR on pituitary samples**. The pituitaries from five male or female fish were pooled for each sample and cultured ex vivo in Leibovitz's L-15 medium supplemented with 5% charcoal-stripped fetal bovine serum as described elsewhere[51]. The medium was further supplemented with $10^{-5}$ M medaka Pacap or Vip, or vehicle alone. After 24 h of culture at 25 °C, the pituitaries were processed for total RNA extraction and cDNA synthesis by using the RNeasy Plus Micro Kit (Qiagen) and SuperScript VILO cDNA Synthesis Kit (Thermo Fisher Scientific), respectively. Real-time PCR was performed on the LightCycler 480 System II using LightCycler 480 SYBR Green I Master (Roche Diagnostics). The cycle conditions were: 95 °C for 5 min followed by 45 cycles of 95 °C for 10 sec, 60 °C for 10 sec and 72 °C for 10 sec. Melting curve analysis was performed as described above. Data were normalized to the geometric mean of three reference genes, *actb*, *gapdh* (encoding glyceraldehyde-3-phosphate dehydrogenase; National BioResource Project (NBRP) Medaka clone ID olbrno9_f15) and *rpl13* (encoding ribosomal protein L13; NBRP Medaka clone ID olovano51_b24), according to Vandesompele et al.[63]. The primers used for real-time PCR are listed in Supplementary Table 3.

**Statistics and reproducibility**. Quantitative data were expressed as means with error bars representing standard error of the mean. Individual data points were also plotted to give a better indication of the underlying distribution. In real-time PCR analysis, the values for control samples (male brain for *vip*; female brain for *adcyap1* and pituitary hormone gene) were arbitrarily set to 1 and the other values were adjusted accordingly.

Statistical analyses were performed by using GraphPad Prism (ver. 8; GraphPad Software, San Diego, CA, USA). Data were compared between two groups by using unpaired two-tailed Student's *t*-test. Welch's correction was applied if the F-test indicated that the variance differed significantly between groups. Bonferroni-Dunn correction was applied for multiple comparisons between two groups. Data were compared among more than two groups by using one-way analysis of variance (ANOVA) followed by Bonferroni's post-hoc test. Homogeneity of variance was verified for all datasets by using Brown-Forsythe test. Two-way ANOVA followed by either Bonferroni's (for comparisons among experimental groups) or Dunnett's (for comparisons of experimental versus control groups) post-hoc test was used for analyses of *adcyap1/vip* expression throughout the diurnal/reproductive cycle and during growth/sexual maturation, ER expression in *adcyap1* and *vip* neurons and pituitary hormone gene expression in response to Pacap/Vip supplementation. No test for outliers was performed and all data were included in the statistical analyses.

**Reporting summary**. Further information on research design is available in the Nature Research Reporting Summary linked to this article.

## Data availability
The medaka *adcyap1* and *vip* cDNA sequences have been deposited in GenBank under accession numbers LC579549 and LC579550, respectively. Source data for all graphs are provided in Supplementary Data 1. The original uncropped images of northern blots are shown in Supplementary Fig. 6. All other data supporting the findings of this study are available within the article and its supplementary information or from the corresponding author upon reasonable request.

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

## Acknowledgements

We thank Dr. Mikoto Nakajo for technical advice on neuronal tracing and Akira Hirata for assistance with medaka husbandry. This work was supported by the Ministry of Education, Culture, Sports, Science, and Technology (MEXT) in Japan and the Japan Society for the Promotion of Science (JSPS) (MEXT/JSPS grant numbers 17J08702 (to J.Y.), 16H04979, 17H06429 and 19H03044 (to K.O.)).

## Author contributions

J.Y. and K.O. designed the research; J.Y., Y.N., T.F., D.K. and K.O. performed the research and analyzed/interpreted the data; and J.Y. and K.O. wrote the manuscript.

## Competing interests

The authors declare no competing interests.
