## [Peer Review File · Communications Biology]

Reviewers' comments:

Reviewer #1 (Remarks to the Author):

The manuscript by Yamashita and Okubo described their recent study using medaka as a model to (i) demonstrate the sexual bias of PACAP and VIP neurons located in the POA of the telencephalon projecting into the pituitary, and (ii) elucidate the underlying mechanisms as well as the functional role of the two neuropeptides in regulating pituitary hormones in both the male and female fish. Using subtraction hybridization coupled with real-time PCR, sexual dimorphism on brain expression of PACAP (with biased expression in the male) and VIP (with biased expression in the female) was noted and the differential expression of the two gene targets was mapped to the respective neurons in the PMp region of the POA and confirmed to be projecting into the pituitary by ISH & retrograde tracking techniques. Using sex-reverse XX male and XY female together with the traditional gonadectomy followed by steroid replacement in normal male & female, the opposite bias of PACAP and VIP neurons in the PMp region was confirmed to be caused by the differential production of estrogen at the gonadal level but not by the genetic effect related to the XX/XY chromosomes. Since the pituitary in medaka was confirmed to be innervated by PACAP & VIP neurons from the PMp region and some of these neurons were also shown to express different types of estrogen receptors, the pituitary actions of the two neuropeptides were also examined along with the molecular cloning, functional characterization & pituitary expression of the six subtypes of PACAP/VIP receptors, including the *Adcyapr1a* & *1b*, *Vipr1a* & *1b* and *Vipr2a* & *2b*. In this case, the different subtypes of PACAP/VIP receptors were found to have different selectivity for the two ligands and exhibited a differential pattern of expression at the pituitary level. Of note, the *Vipr2b* receptor subtype was found to be highly expressed in the male but not the female pituitary. Similarly, using *ex vivo* pituitary culture, the two neuropeptides were also confirmed to have differential effects on gene expression of different pituitary hormones, and some of the hormone responses were found to be sexual dimorphic. These findings, as a whole, demonstrate for the first time that the sexual bias of PACAP & VIP neurons located in the POA area of the brain caused by the difference in estrogen milieu between the male and female along with different subtypes of PACAP/VIP receptors expressed in different areas of the pituitary may lead to the differential regulation of pituitary hormones in the opposite sex in a fish model.

The manuscript is well-written and covers the novel findings related to sexual dimorphism in the brain-pituitary axis in a fish model which is widely used in the field. The study involved a large amount of works, and the experiments were carefully designed & highly logical and the data presented are of high quality, I do not have any questions related to the results & data interpretation. I have only two minor comments with the objective to further improve on the discussion and they are listed as follows:

(1) In the summary section of the discussion (L394-396), the statement of "the differential use of neuropeptides for brain-pituitary signaling most probably functions to prevent sex difference in pituitary hormones by compensating for differences in the steroid milieu" may need to be modified, as the results presented related to (i) sexual bias of PACAP & VIP neurons in the POA, (ii) sexual difference of *Vipr2a* expressed at the pituitary level, and (iii) sexual dimorphism observed for PACAP & VIP on some of the pituitary hormones clearly shown that some of the hormone responses not to mentioned the corresponding basal level of pituitary hormone expression are sexual dimorphic. These is consistent with the reports in other fish species on the difference in body growth & adiposity and/or differential maturation rate observed between the two sexes. I suppose the summary statement for the discussion should be more align with the physiological relevance of the sexual dimorphism in the brain-pituitary axis rather than the "potential compensation" at the pituitary level.

(2) Given that the PAC1 & VPAC receptors are commonly coupled with cAMP and calcium signals, which are linked with the stimulatory responses for hormone studies, the inhibitory effects on FSHb & POMC gene expression by PACAP & VIP with concurrent stimulation on other pituitary hormones (e.g., LHb, SL & PRL mRNA) are intriguing, especially in terms of the mechanistic aspect of signal transduction at the cellular level. Since the *ex vivo* culture was based on whole pituitaries pooled from

different fish, would it be possible that some of the inhibitory actions was not a direct effect on pituitary cells but mediated by neurons/nerve terminals still present in the pituitary ? (i.e., a situation similar to VIP actions acting indirectly via other neurons within the hypothalamus in rodents.) Some discussion on the "functional difference" between the ex vivo pituitary culture versus dispersed pituitary cell culture may help to cover the other possibility in data interpretation.

Reviewer #2 (Remarks to the Author):

This study investigated the expressions and functions of pituitary adenylate cyclase activating polypeptide (pacap) and vasoactive intestinal peptide (vip) in the brains of male and female fishes. The expressions of these two genes were determined by in situ hybridization and RT-PCR, and their sexually dimorphic expressions were found in not all but some particular nuclei. The gene expressions are likely to be regulated (activated for vip and suppressed for pacap) by estrogen in the female: estrogen rather than sex chromosomes potentially produces female-specific and thus different from male gene expression. The Pacap and Vip display pleiotropic effects on the pituitary functions possibly through binding to the receptors.

All the experiments seem to be done quite precisely and thus the results presented in this manuscript are reliable. This study may have the potential to be published. Unfortunately, however, the reviewer had an impression that the results presented in the manuscript are mostly descriptive.

Specific comments

1, The fish displays many sexually dimorphic features, such as organ/tissue/cellular functions, morphologies, behaviors, metabolisms, etc., although the mechanisms to induce the differences remain unclear. Under this current status, what is expected for this study is to elucidate the mechanism to establish the particular sexually dimorphic features. Although it was demonstrated by the current study that the expression levels of Pacap and Vip are different between the two sexes in multiple regions of the brain, it remains unclear whether any of the sexually dimorphic features are established and/or controlled by the sexually dimorphic expression of the peptide hormones and moreover how the sexually dimorphic expression of the peptides causes the sexually dimorphic features.

Does the pituitary of the fish display sexually dimorphic functions? The expression levels of the hormones in vivo are different between the two sexes? Fshb and lhb gene expressions in the female are constant or variable during follicle cycle? If the expressions of gh and pomc are different between the two sexes, what is induced by the differential expressions of these hormones?

2, If the pituitary functions of the fish are different between the two sexes, are they attributable to the sexually dimorphic expressions of Pacap and Vip? In vivo but not in vitro evidence could strengthen this study.

3, How much are the concentrations of Adcyap1 and Vip around the pituitary? If the concentration is lower than 10^{-7} , Vipr2a and b are functional in vivo? In vivo concentrations of the peptides are needed to evaluate the results indicated in Fig. 5. Although it might be impossible to determine those directly, are there indirect methods for the estimation?

4, In the study shown in Fig. 6, 10^{-5} M Adcyap1 and Vip were added to the culture media. When lower concentrations of them are added, results different from those shown in Fig. 6 could be obtained?

5, The estradiol concentration in the blood or ovary is similar between XX female and XY female? Is it constant in the female? Ovarian cycle does not affect it?

Minor comments

1, Why the authors did not determine the concentration of cAMP directly but determined indirectly via CRE-luciferase gene expression? Direct determination is better than indirect one.

2, The colors of the bars in the figures are confusing. In some case, blue and yellow bars indicate female and male, respectively, and in other case they indicate treatments with Vip and Pacap1, but not sexes.

Reviewer #3 (Remarks to the Author):

The manuscript describes the discovery of a sexual dimorphic nucleus in the brain of the Japanese ricefish, probably the first description of a sexually dimorphic nucleus in fish. According to the authors, this hypophysiotropic nucleus secretes PACAP (in males) or VIP (in females) to control the release of various pituitary hormones. The manuscript is clearly written and research is very well executed and presented. My main concern regards the interpretation of the results and the conclusion of the manuscript: Since both PACAP and VIP bind and activate the same receptors, it is unclear what is the physiological relevance of the sexually dimorphic expression of the peptides. The author's conclusion that the dimorphism acts to prevent differences in hormone secretion between the sexes does not make sense in evolutionary terms and makes it clear that this nucleus does not mediate the physiological difference between males and females.

Some specific comments:

Line 189: Can the authors quantify the percentage of cells expressing each estrogen receptor?

Line 203: What percentage of the cells were labeled by the neurobiotin? Were only PMP neurons labeled by the neurobiotin or did the authors observe retrograde labeling in PACAP/VIP neurons located in other areas?

Line 211: The word "primarily" means that most of the neurons project to the pituitary. Is that the case?

Lines 237-341: Since the authors do not co-localize receptors onto specific cell types in the pituitary, they should at least try to match the receptor expression patterns to the known expression locations of pituitary hormone cells.

Line 267: reconsider the term hypophysiotropic functions. So far in the results the authors have shown hypophysiotropic projections and expression of receptors. Functions is the next stage.

Lines 265-294: Do receptor locations match with the hormones most affected. For example are there receptors (which ones?) in the RPD, where prolactin cells are located?

Lines 265-294: Considering the much stronger expression of vip2 receptor in males, one would expect that the effect on male pituitaries (at least for hormones expressed in the posterior part) would be much stronger. There does not seem to be such a response.

Lines 265-294: Why should the same peptides have stimulatory effects on the expression of some hormones and inhibitory effects on others? Does this match known signal-transduction cascades of the different receptors?

Line 305: Are there other examples of males and females using different neuropeptides for equivalent signaling functions?

Line 308: Is it possible that these peptides are co-expressed with other hypophysiotropic peptides with more established pituitary functions?

Line 395: As mentioned earlier, the conclusion does not make much evolutionary sense. Is it possible that considering the direct innervation of the pituitary, the neurons target different pituitary cells in males and females? Or that neurons from this nucleus target other brain areas which differentially express the receptors? Or that the different affinities of the receptors for the ligands induce a

differential response?

Lines 454-462: Only a single reference gene was used for all real-time quantifications? Can the authors show that its levels were not affected by treatments?

Line 527: The reference states 30-60 minutes for the neurobiotin retrograde labeling. Was 5 minutes enough in the current study?

Fig 1D: Could the authors mark the spawning hour with an arrow?

Fig 1E: Could the authors mark age of puberty with an arrow?

Line 739: I think "in" should be "on".

Fig 2: Nice anatomical maps like in the sup figs will help reader's orientation.

Fig 4C: Is that a male or a female? Mention at least in the legend.

Fig 5C: Some annotation would be useful: RPD, PPD, PI etc., Maybe also a scheme of the medaka pituitary detailing the known location of the different hormone-expressing cells.

Fig 6: Some kind of summary scheme for the whole paper (like a graphical abstract) would be useful.

Responses to the reviewers

First of all, we would like to thank the three anonymous reviewers for the positive evaluation of our work and the constructive comments and suggestions, which significantly helped improve the manuscript. We greatly appreciate their time and effort. We have considered their comments and suggestions carefully and made revisions accordingly. In the revised manuscript, revised portions are indicated in **red letters**. In this document, the original text is indicated in **blue letters** and the revised text is in **red letters**. All line numbers in this document refer to those in the revised manuscript.

Reviewer #1

Reviewer #1's overall comment:

The manuscript by Yamashita and Okubo described their recent study using medaka as a model to (i) demonstrate the sexual bias of PACAP and VIP neurons located in the POA of the telencephalon projecting into the pituitary, and (ii) elucidate the underlying mechanisms as well as the functional role of the two neuropeptides in regulating pituitary hormones in both the male and female fish. Using subtraction hybridization coupled with real-time PCR, sexual dimorphism on brain expression of PACAP (with biased expression in the male) and VIP (with biased expression in the female) was noted and the differential expression of the two gene targets was mapped to the respective neurons in the PMp region of the POA and confirmed to be projecting into the pituitary by ISH & retrograde tracking techniques. Using sex-reverse XX male and XY female together with the traditional gonadectomy followed by steroid replacement in normal male & female, the opposite bias of PACAP and VIP neurons in the PMp region was confirmed to be caused by the differential production of estrogen at the gonadal level but not by the genetic effect related to the XX/XY chromosomes. Since the pituitary in medaka was confirmed to be innervated by PACAP & VIP neurons from the PMp region and some of these neurons were also shown to express different types of estrogen receptors, the pituitary actions of the two neuropeptides were also examined along with the molecular cloning, functional characterization & pituitary expression of the six subtypes of PACAP/VIP receptors, including the Adcyap1a & 1b, Vipr1a & 1b and Vipr2a & 2b. In this case, the different subtypes of PACAP/VIP receptors were found to have different selectivity for the two ligands and exhibited a differential pattern of expression at the pituitary level. Of note, the Vipr2b receptor subtype was found to be highly expressed in the male but not the female pituitary. Similarly, using ex vivo pituitary culture, the two neuropeptides were also confirmed to have differential effects on gene expression of different pituitary hormones, and some of the hormone responses were found to be sexual dimorphic. These findings, as a whole, demonstrate for the first time that the sexual bias of PACAP & VIP neurons located in the POA area of the brain caused by the difference in estrogen milieu between the male and female along with different subtypes of PACAP/VIP receptors expressed in different areas of the pituitary may lead to the differential regulation of pituitary hormones in the opposite sex in a fish model.

The manuscript is well-written and covers the novel findings related to sexual dimorphism in the brain-pituitary axis in a fish model which is widely used in the field. The study involved a large amount of works, and the experiments were carefully designed & highly logical and the data presented are of high quality, I do not have any questions related to the results & data interpretation. I have only two minor comments with the objective to further improve on the discussion and they are listed as follows:

Response to reviewer #1's overall comment:

We thank the reviewer for this very positive evaluation of our work and greatly appreciate their helpful comments for improving the manuscript.

Reviewer #1's specific comment 1:

(1) In the summary section of the discussion (L394-396), the statement of “the differential use of neuropeptides for brain-pituitary signaling most probably functions to prevent sex difference in pituitary hormones by compensating for differences in the steroid milieu” may need to be modified, as the results presented related to (i) sexual bias of PACAP & VIP neurons in the POA, (ii) sexual difference of *Vipr2a* expressed at the pituitary level, and (iii) sexual dimorphism observed for PACAP & VIP on some of the pituitary hormones clearly shown that some of the hormone responses not to mentioned the corresponding basal level of pituitary hormone expression are sexual dimorphic. These is consistent with the reports in other fish species on the difference in body growth & adiposity and/or differential maturation rate observed between the two sexes. I suppose the summary statement for the discussion should be more align with the physiological relevance of the sexual dimorphism in the brain-pituitary axis rather than the “potential compensation” at the pituitary level.

Response to reviewer #1's specific comment 1:

We agree with this comment, but would like to add that it is becoming increasingly accepted that not all sex differences in the brain function to establish sexually dimorphic features and that some of them function to reduce sex differences caused by other factors (*e.g.*, De Vries, 2004, *Endocrinology*, 145:1063–1068; Arnold, 2014, *Exp Neurol*, 259:2–9; McCarthy, 2016, *Philos Transact Royal Soc B Biol Sci*, 371:20150106). The sex difference in *Pacp* and *Vip* expression identified in the present study is considered a typical example of such sex differences.

Line 427: By considering this information and the above comment from the reviewer, we have revised the summary statement “**This differential use of neuropeptides for brain–pituitary signaling most probably functions to prevent sex differences in pituitary hormones, by compensating for differences in the steroid milieu.**” to “**It is becoming increasingly accepted that not all sex differences in the brain cause sex differences in physiological and behavioral traits and that some of them serve to prevent sex differences in these traits (De Vries, 2004; Arnold, 2014; McCarthy, 2016). The differential expression of neuropeptides for brain–pituitary signaling identified in the present study is considered a typical example of such sex differences and most probably serves to prevent undesirable sex differences in gametogenesis, ion and water transport, and skin coloration. The differential use of neuropeptides that are oppositely regulated by estrogen may represent a regulatory mechanism that has evolved to ensure these pituitary functions regardless of the estrogen milieu.**”. We hope that the reviewer will be pleased with this revision.

Line 427: With the above change, “**sex steroid milieu**” has been replaced with “**estrogen milieu**”.

Reference list: The following two references, which are cited in the above text, have been added.

De Vries GJ (2004) Sex differences in adult and developing brains: compensation, compensation, compensation. *Endocrinology* 145:1063–1068. doi:10.1210/en.2003-1504.

Arnold AP (2014) Conceptual frameworks and mouse models for studying sex differences in physiology and disease: why compensation changes the game. *Exp Neurol* 259:2–9. doi:10.1016/j.expneurol.2014.01.021.

Reviewer #1's specific comment 2:

(2) Given that the PAC1 & VPAC receptors are commonly coupled with cAMP and calcium signals, which are linked with the stimulatory responses for hormone studies, the inhibitory effects on FSHb & POMC gene expression by PACAP & VIP with concurrent stimulation on other pituitary hormones (e.g., LHb, SL & PRL mRNA) are intriguing, especially in terms of the mechanistic aspect of signal transduction at the cellular level. Since the ex vivo culture was based on whole pituitaries pooled from different fish, would it be possible that some of the inhibitory actions was not a direct effect on pituitary cells but mediated by neurons/nerve terminals still present in the pituitary ? (i.e., a situation similar to VIP actions acting indirectly via other neurons within the hypothalamus in rodents.) Some discussion on the “functional difference” between the ex vivo pituitary culture versus dispersed pituitary cell culture may help to cover the other possibility in data interpretation.

Response to reviewer #1's specific comment 2:

We agree that the mechanism by which Pacap and Vip inhibit the expression of *fshb* and *pomc* is certainly an interesting question. Although no information is available for the inhibitory effect on *pomc* other than our present study, there have been several published reports on *fshb*. Notably, it has been shown in rats that PACAP is effective in inhibiting *Fshb* expression even when administered to dispersed pituitary cells, indicating a direct action of PACAP on gonadotrophs (Köves *et al.*, 2020, *Front Endocrinol*, 11:88; Winters and Moore, 2020, *Mol Cell Endocrinol*, 518:110912). Furthermore, this effect has been shown to be mediated by increased expression of follistatin. It thus seems possible that Pacap may have a similar mechanism of action in medaka.

Line 407: To provide this information, we have added the following text to the Discussion section: “**Studies on dispersed rat pituitary cells indicate that PACAP acts directly on gonadotrophs to inhibit *Fshb* expression and that this effect is mediated by increased expression of follistatin (Köves *et al.*, 2020; Winters and Moore, 2020). Possibly, Pacap may have a similar mechanism of action in medaka.**”.

Line 411: With the addition of this text, “**this effect**” has been replaced with “**the inhibitory effect on *fshb***”.

Reviewer #2

Reviewer #2's overall comment:

This study investigated the expressions and functions of pituitary adenylate cyclase activating polypeptide (pacap) and vasoactive intestinal peptide (vip) in the brains of male and female fishes. The expressions of these two genes were determined by in situ hybridization and RT-PCR, and their sexually dimorphic expressions were found in not all but some particular nuclei. The gene expressions are likely to be regulated (activated for vip and suppressed for pacap) by estrogen in the female: estrogen rather than sex chromosomes potentially produces female-specific and thus different from male gene expression. The Pacap and Vip display pleiotropic effects on the pituitary functions possibly through binding to the receptors.

All the experiments seem to be done quite precisely and thus the results presented in this manuscript are reliable. This study may have the potential to be published. Unfortunately, however, the reviewer had

an impression that the results presented in the manuscript are mostly descriptive.

Response to reviewer #2's overall comment:

We thank the reviewer for their constructive comments, which we found very useful to improve the manuscript. Below is a point-by-point response to the comments.

Reviewer #2's specific comment 1:

1, The fish displays many sexually dimorphic features, such as organ/tissue/cellular functions, morphologies, behaviors, metabolisms, etc., although the mechanisms to induce the differences remain unclear. Under this current status, what is expected for this study is to elucidate the mechanism to establish the particular sexually dimorphic features. Although it was demonstrated by the current study that the expression levels of Pacap and Vip are different between the two sexes in multiple regions of the brain, it remains unclear whether any of the sexually dimorphic features are established and/or controlled by the sexually dimorphic expression of the peptide hormones and moreover how the sexually dimorphic expression of the peptides causes the sexually dimorphic features.

Does the pituitary of the fish display sexually dimorphic functions? The expression levels of the hormones in vivo are different between the two sexes? Fshb and lhb gene expressions in the female are constant or variable during follicle cycle? If the expressions of gh and pomc are different between the two sexes, what is induced by the differential expressions of these hormones?

Response to reviewer #2's specific comment 1:

We previously showed that the expression of many pituitary hormones is sexually dimorphic in medaka (Kawabata-Sakata *et al.*, 2020, *Proc Royal Soc B*, 287:20200713), suggesting sexual dimorphism in various pituitary functions, including gametogenesis, ion and water transport, and skin coloration. In the present study, we conclude that the sexual dimorphism in Pacap and Vip expression in the POA probably serves to reduce sex differences in the expression of pituitary hormones and hence pituitary functions. Here, we would like to mention that it is becoming increasingly accepted that not all sex differences in the brain function to establish sexually dimorphic features and that some of them function to reduce sex differences caused by other factors (*e.g.*, De Vries, 2004, *Endocrinology*, 145:1063–1068; Arnold, 2014, *Exp Neurol*, 259:2–9; McCarthy, 2016, *Philos Transact Royal Soc B Biol Sci*, 371:20150106). The sex difference in Pacap and Vip expression identified in the present study is considered a typical example of such sex differences.

Line 427: To include these arguments in the revised manuscript, the summary sentence “**This differential use of neuropeptides for brain–pituitary signaling most probably functions to prevent sex differences in pituitary hormones, by compensating for differences in the steroid milieu.**” has been modified and now reads “**It is becoming increasingly accepted that not all sex differences in the brain cause sex differences in physiological and behavioral traits and that some of them serve to prevent sex differences in these traits (De Vries, 2004; Arnold, 2014; McCarthy, 2016). The differential expression of neuropeptides for brain–pituitary signaling identified in the present study is considered a typical example of such sex differences and most probably serves to prevent undesirable sex differences in gametogenesis, ion and water transport, and skin coloration. The differential use of neuropeptides that are oppositely regulated by estrogen may represent a regulatory mechanism that has evolved to ensure these pituitary functions regardless of the estrogen milieu.**”.

Line 427: With the above change, “sex steroid milieu” has been replaced with “estrogen milieu”.

Reference list: The following two references, which are cited in the above text, have been added.

De Vries GJ (2004) Sex differences in adult and developing brains: compensation, compensation, compensation. *Endocrinology* 145:1063–1068. doi:10.1210/en.2003-1504.

Arnold AP (2014) Conceptual frameworks and mouse models for studying sex differences in physiology and disease: why compensation changes the game. *Exp Neurol* 259:2–9. doi:10.1016/j.expneurol.2014.01.021.

On the other hand, as the reviewer notes, we identified sex differences in Pacap and Vip expression in several brain regions, but we do not yet know what physiological and behavioral traits are associated with the sex differences outside of the POA. We also consider it necessary to clarify this issue, and to this end, we are currently planning to generate Pacap and Vip mutant medaka and analyze their phenotypes, including a variety of behaviors. However, this will take a lot of time, and we hope to report the results in a forthcoming paper.

Line 395: As the reviewer points out, the basal expression of *gh* is higher in female pituitaries, while that of *pomc* is higher in male pituitaries. The female-biased expression of *gh* may be related to the fact that female medaka have larger body size than males. However, since we have already reported this in a previous study (Kawabata-Sakata *et al.*, 2020, *Proc Royal Soc B* 287:20200713) and *gh* expression was not affected by Pacap and Vip administration, we did not specifically discuss the significance of the female-biased expression of *gh*. On the other hand, we referred to the higher basal expression of *pomc* in males as follows in the Discussion section: “*Pomc* is post-translationally processed in the intermediate lobe to yield α -melanocyte-stimulating hormone, whose primary function in teleosts is to regulate skin coloration (Takahashi *et al.*, 2009). Although this function is important regardless of sex, the expression of *pomc* is male-biased (this study and Kawabata-Sakata *et al.*, 2020). Similar to *sl*, the male-dominant action of Pacap may serve to reduce the sex difference in *pomc* expression, which would otherwise be even larger.”. In response to the above comment from the reviewer, this text has been modified and now reads “*Pomc* is post-translationally processed in the PI to yield α -melanocyte-stimulating hormone, whose primary function in teleosts is to regulate skin coloration (Takahashi *et al.*, 2009). Consistent with the fact that male medaka generally have darker and more vivid skin color than females, the basal expression of *pomc* is male-biased (this study and Kawabata-Sakata *et al.*, 2020). Similar to *sl*, the male-dominant action of Pacap may serve to reduce the sex difference in *pomc* expression, which would otherwise be even larger. In other words, Pacap may prevent extreme skin darkening in males by suppressing *pomc*. Medaka prefer mates with skin color similar to their own and avoid mates with skin color that differ greatly from their own, suggesting that pre-mating reproductive isolation is occurring between strains of medaka with different skin colors (Ikawa *et al.*, 2017). The differential use of Pacap and Vip may contribute to this process by reducing sex differences in skin color through the regulation of *sl* and *pomc*.”.

Reference list: The following reference, which is newly cited in the above text, has been added.

Ikawa M, Ohya E, Shimada H, Kamijo M, Fukamachi S (2017) Establishment and maintenance of sexual preferences that cause a reproductive isolation between medaka strains in close association. *Biol Open* 6:244–251. doi:10.1242/bio.022285.

Changes in the expression of *fshb* and *lhb* during the reproductive cycle of female medaka have been

reported in a previous study (Karigo *et al.*, 2012, *Endocrinology*, 153:3394–3404). Medaka have a 24 hour reproductive cycle, typically spawning at the beginning of the light phase. The expression of both *fshb* and *lhb* is highest in the late dark phase (5:00) and lowest in the late light phase (13:00–21:00), as shown below.

Karigo *et al.*, 2012

These changes in expression have been explained by the cyclic nature of the action of gonadotropin-releasing hormone 1 (Gnrh1) (Karigo *et al.*, 2012, *Endocrinology*, 153:3394–3404), but it would be interesting for further studies to examine whether the regulation by Pacap and Vip, as found in the present study, is also involved in these changes.

Line 420: To include these arguments in the revised manuscript, the following text has been added to the Discussion section: “The expression levels of both *fshb* and *lhb* have been shown to fluctuate with the diurnal/reproductive cycle of female medaka (Karigo *et al.*, 2012). These fluctuations are attributed to the cyclic nature of the action of gonadotropin-releasing hormone 1 (Karigo *et al.*, 2012), but it may be worthwhile in future studies to assess whether Pacap and Vip are also involved in these fluctuations.”.

Reviewer #2’s specific comment 2:

2, If the pituitary functions of the fish are different between the two sexes, are they attributable to the sexually dimorphic expressions of Pacap and Vip? In vivo but not in vitro evidence could strengthen this study.

Response to reviewer #2’s specific comment 2:

As described above, there are probably sex differences in various pituitary functions, but the results of the present study indicate that Pacap and Vip function to prevent those sex differences rather than establish them. *In vivo* evidence, such as that obtained by using mutant medaka, would certainly strengthen the present findings. However, as mentioned above, it will take a lot of time to obtain such evidence, and we hope to report it in a forthcoming paper.

Reviewer #2’s specific comment 3:

3, How much are the concentrations of Adcyap1 and Vip around the pituitary? If the concentration is lower than 10⁻⁷, Vipr2a and b are functional in vivo? In vivo concentrations of the peptides are needed to evaluate the results indicated in Fig. 5. Although it might be impossible to determine those directly, are there indirect methods for the estimation?

Response to reviewer #2's specific comment 3:

Because the medaka pituitary is extremely small, measuring the concentrations of Pacap and Vip in the pituitary is quite difficult, and almost impossible with the methods available to us. As mentioned in the manuscript (line 206), the anterior pituitary of teleosts does not have a median eminence/hypophyseal portal system and receives direct innervation from preoptic/hypothalamic neurons. This indicates that in teleosts, preoptic/hypothalamic factors act locally at high concentrations around axon terminals and varicosities in the pituitary. Accordingly, even if we could measure the concentrations of Pacap and Vip in the whole pituitary, we would not be able to obtain meaningful values that reflect the local concentrations at which these peptides actually act. Given these facts and in light of this comment, we have instead conducted immunohistochemistry to investigate whether Pacap and Vip are present in high concentrations locally in the pituitary. The results, although not quantitative, showed a dense plexus of axons and varicosities that were intensely immunoreactive for Pacap and Vip in the anterior pituitary, as shown below (the new Figure 6B; representative micrographs showing Pacap and Vip immunoreactivity (green) in the anterior pituitary). This suggests that Pacap and Vip indeed act locally at high concentrations in the pituitary. This is the best we can do, and we would appreciate it if the reviewer would understand this.

The anti-PACAP and VIP antibodies used have been repeatedly shown to recognize teleost Pacap and Vip, respectively, with high specificity (*e.g.*, anti-PACAP: Olsson and Holmgren, 1994, *Cell Tissue Res*, 277:539–547; Wong *et al.*, 1998, *Endocrinology*, 139:3465–3479; Uyttebroek *et al.*, 2013, *Cell Tissue Res*, 354:355–370. anti-VIP: Olsson and Holmgren, 1994, *Cell Tissue Res*, 277:539–547; Finney *et al.*, 2006, *J Comp Neurol*, 495:587–606; Uyttebroek *et al.*, 2010, *J Comp Neurol*, 518:4419–4438; Uyttebroek *et al.*, 2013, *Cell Tissue Res*, 354:355–370). The specificity of these antibodies was further verified by preabsorbing with the synthetic medaka Pacap and Vip polypeptides, which blocked immunodetection, as shown in the micrographs below (these are for review purposes only and are not included in the manuscript).

To include these results, we have revised the manuscript as follows.

Line 213: “In addition, the direct innervation of the anterior pituitary by Pacap- and Vip-containing axons was confirmed by immunohistochemistry, which revealed a dense plexus of axons and varicosities that were intensely immunoreactive for Pacap and Vip in the anterior pituitary (Figure 6B).” has been added.

Line 392: “Although activation of Vipr2b requires rather high concentrations of Pacap, the immunohistochemical detection of a dense plexus of Pacap-containing axons and varicosities in the pituitary suggests that Pacap can act locally at high concentrations in the pituitary to activate Vipr2b.” has been added.

Line 588: “Immunohistochemistry. The pituitary was fixed in 4% PFA and embedded in 5% agarose (Type IX-A; Sigma-Aldrich) supplemented with 20% sucrose. Frozen 20- μ m thick sections were cut in the coronal plane. After blocking with phosphate-buffered saline (PBS) containing 2% normal goat serum, the sections were incubated overnight at 4°C with anti-PACAP antibody (RRID: AB_519166; Peninsula Laboratories, San Carlos, CA, USA) or anti-VIP antibody (RRID: AB_572270; ImmunoStar, Hudson, WI, USA) diluted at 1:500 or 1:1000, respectively, in PBS containing 2% normal goat serum, 0.1% bovine serum albumin, and 0.02% keyhole limpet hemocyanin. The sections were then reacted overnight at 4°C with Alexa Fluor 488-conjugated goat anti-rabbit IgG (Thermo Fisher Scientific) diluted 1:1000 in PBS. The excitation and emission wavelengths for Alexa Fluor 488 were 488 nm and 495–545 nm, respectively. The anti-PACAP and VIP antibodies used have been shown to recognize teleost Pacap and Vip, respectively, with high specificity (*e.g.*, Olsson and Holmgren, 1994; Wong *et al.*, 1998; Finney *et al.*, 2006; Uyttebroek *et al.*, 2010; 2013). The specificity of these antibodies was further verified by preabsorbing with the synthetic medaka Pacap and Vip polypeptides (see below), which blocked immunodetection.” has been added.

Figure 6: The immunohistochemical images shown above have been included in the revised version of the paper as Figure 6B (these images have been added to the original Figure 4, but this caused the size of the figure to exceed one page; accordingly, we have divided it into two figures (new Figures 5 and 6)).

Line 919: The legend for the new Figure 6 reads: “*adcyp1* and *vip* are expressed in the same population of POA neurons that project to the pituitary. (A) Retrograde labeling of *adcyp1*- and *vip*-expressing neurons in the PMp after injection of the neuronal tracer Neurobiotin into the anterior pituitary. Left and middle panels show images of, respectively, *adcyp1/vip* expression (green) and Neurobiotin labeling (magenta) in the same section; right panel shows the merged image with nuclear counterstaining (blue). Arrowheads indicate representative *adcyp1/vip*-expressing neurons labeled with Neurobiotin. Representative images of *adcyp1*- and *vip*-expressing neurons in a male and female, respectively, are presented. (B) Immunohistochemical detection of Pacap- and Vip-containing axons and varicosities (green) in the male and female anterior pituitary. Blue color indicates nuclear counterstaining. (C) Intermingled and partially overlapping distribution of *adcyp1*- and *vip*-expressing neurons in the male and female PMp. Representative micrographs taken at two different levels along the anterior–posterior axis are shown. Left and middle panels show images of, respectively, *adcyp1* (green) and *vip* (magenta) expression in the same section; right panel shows the merged image with nuclear counterstaining (blue).

Arrowheads indicate representative neurons co-expressing *adcyap1* and *vip*. Scale bars represent 20 μm .”.

Lines 212 and 221: With this change, “Figure 4C” and “Figure 4D” have been changed to “Figure 6A” and “Figure 6C”, respectively.

Reference list: The following references, which are cited in the above text, have been added.

Finney JL, Robertson GN, McGee CA, Smith FM, Croll RP (2006) Structure and autonomic innervation of the swim bladder in the zebrafish (*Danio rerio*). *J Comp Neurol* 495:587–606. doi:10.1002/cne.20948.

Olsson C, Holmgren S (1994) Distribution of PACAP (pituitary adenylate cyclase-activating polypeptide)-like and helospectin-like peptides in the teleost gut. *Cell Tissue Res* 277:539–547. doi:10.1007/BF00300227.

Uyttebroek L, Shepherd IT, Harrisson F, Hubens G, Blust R, Timmermans JP, Van Nassauw L (2010) Neurochemical coding of enteric neurons in adult and embryonic zebrafish (*Danio rerio*). *J Comp Neurol* 518:4419–4438. doi:10.1002/cne.22464.

Uyttebroek L, Shepherd IT, Hubens G, Timmermans JP, Van Nassauw L (2013) Expression of neuropeptides and anoctamin 1 in the embryonic and adult zebrafish intestine, revealing neuronal subpopulations and ICC-like cells. *Cell Tissue Res* 354:355–370. doi:10.1007/s00441-013-1685-8.

Wong AO, Leung MY, Shea WL, Tse LY, Chang JP, Chow BK (1998) Hypophysiotropic action of pituitary adenylate cyclase-activating polypeptide (PACAP) in the goldfish: immunohistochemical demonstration of PACAP in the pituitary, PACAP stimulation of growth hormone release from pituitary cells, and molecular cloning of pituitary type I PACAP receptor. *Endocrinology* 139:3465–3479. doi:10.1210/endo.139.8.6145.

Reviewer #2’s specific comment 4:

4, In the study shown in Fig. 6, 10⁻⁵ M Adcyap1 and Vip were added to the culture media. When lower concentrations of them are added, results different from those shown in Fig. 6 could be obtained?

Response to reviewer #2’s specific comment 4:

Line 385: Because we were uncertain whether Pacap and Vip had any effect on pituitary hormones, we administered these peptides at a high concentration to maximize their effect. Considering the EC₅₀ of their receptors, a different result may be obtained if these peptides are administered at a lower concentration. In relation to this, we had included the following argument in the Discussion section: “Pacap and Vip were equally effective in stimulating *sl* and *prl* expression, probably because both neuropeptides were administered at a relatively high concentration. However, the potency of Pacap was more than 100-fold higher than that of Vip in activating both *Adcyap1r1a* and *Adcyap1r1b* in medaka, as has been reported in other species (Vaudry *et al.*, 2009; Harmar *et al.*, 2012), and thus is presumably more effective than Vip at lower concentrations.”. In response to this comment, we have added the following sentence after this argument: “Additional testing with lower concentrations of Pacap and Vip will be necessary to verify this presumption.”.

Reviewer #2’s specific comment 5:

5, The estradiol concentration in the blood or ovary is similar between XX female and XY female? Is it constant in the female? Ovarian cycle does not affect it?

Response to reviewer #2’s specific comment 5:

Although there have been no studies comparing blood E2 levels in XX and XY females, two previous studies investigated the changes in blood E2 levels during the reproductive cycle in female medaka (Soyano *et al.*, 1993, *Fish Physiol Biochem*, 11:265–272; Kayo *et al.*, 2020, *Gen Comp Endocrinol*, 285:113272). Both studies show that blood E2 levels fluctuate with the reproductive cycle and are lowest in the middle of the light phase, as shown below.

Soyano *et al.*, 1993

Fig. 1. Diurnal changes in plasma concentrations of estradiol-17 β in female medaka.

Kayo *et al.*, 2020

Line 337: In the present study, we show that the sex differences in the expression of both *adcyp1* and *vip* do not reach significance in the late light phase (please see Figure 1D and lines 101–111). We did not discuss this result specifically, but given the fluctuation in E2 levels described above, we now assume that this reduction in sex differences may be due to a decrease in E2 levels in females. Therefore, we have added the following argument to the Discussion section: “Consistent with this observation, we also found that sex differences in *adcyp1* and *vip* expression decrease after the mid-light phase, when blood estrogen levels in females are at their lowest (Soyano *et al.*, 1993; Kayo *et al.*, 2020)”. We deeply appreciate the discerning comment from this reviewer.

Reference list: The following two references, which are cited in the above text, have been added.

Kayo D, Oka Y, Kanda S (2020) Examination of methods for manipulating serum 17 β -estradiol (E2) levels by analysis of blood E2 concentration in medaka (*Oryzias latipes*). *Gen Comp Endocrinol* 285:113272. doi:10.1016/j.ygcen.2019.113272.

Soyano K, Saito T, Nagae M, Yamauchi K (1993) Effects of thyroid hormone on gonadotropin-induced steroid production in medaka, *Oryzias latipes*, ovarian follicles. *Fish Physiol Biochem* 11:265–272. doi:10.1007/BF00004574.

Reviewer #2's minor comment 1:

1, Why the authors did not determine the concentration of cAMP directly but determined indirectly via CRE-luciferase gene expression? Direct determination is better than indirect one.

Response to reviewer #2's minor comment 1:

It is true that the direct detection method may be superior, but the indirect detection method using the pGL4.29 luciferase vector has been well validated and is widely used (*e.g.*, please see Cheng *et al.*, 2010, *Curr Chem Genomics*, 4:84–91; <https://www.promege.jp/-/media/files/resources/cell-notes/cn017/enhanced-response-dynamics-for-transcription-analysis-using-new-pgl4-luciferase->

reporter-vectors.pdf?la=en). We see no problem in using this method.

Reviewer #2's minor comment 2:

2, The colors of the bars in the figures are confusing. In some case, blue and yellow bars indicate female and male, respectively, and in other case they indicate treatments with Vip and Pacap1, but not sexes.

Response to reviewer #2's minor comment 2:

Figures 4D, 4F, and 8: We agree with this comment and have changed the color of the columns in these figures. In the revised version of the paper, all columns for data obtained from males are shown in yellow, and all columns for females are shown in blue.

Reviewer #3

Reviewer #3's overall comment:

The manuscript describes the discovery of a sexual dimorphic nucleus in the brain of the Japanese ricefish, probably the first description of a sexually dimorphic nucleus in fish. According to the authors, this hypophysiotropic nucleus secretes PACAP (in males) or VIP (in females) to control the release of various pituitary hormones. The manuscript is clearly written and research is very well executed and presented. My main concern regards the interpretation of the results and the conclusion of the manuscript: Since both PACAP and VIP bind and activate the same receptors, it is unclear what is the physiological relevance of the sexually dimorphic expression of the peptides. The author's conclusion that the dimorphism acts to prevent differences in hormone secretion between the sexes does not make sense in evolutionary terms and makes it clear that this nucleus does not mediate the physiological difference between males and females.

Response to reviewer #3's overall comment:

We sincerely thank the reviewer for their positive comments and constructive suggestions. We understand their concern regarding our conclusion. In the present study, we conclude that the sexually dimorphic expression of Pacap and Vip in the POA probably serves to reduce sex differences in the expression of pituitary hormones and hence pituitary functions. Here, we would like to add a few words to this conclusion. If only Vip regulated pituitary hormones and Pacap had no such function, for example, *pomc*, whose basal expression level is higher in males, would be expressed even more highly in males. This is because, in reality, Pacap suppresses *pomc* expression in a male-specific manner. Conversely, if Pacap were the only regulator of pituitary hormones, for example, *fshb*, whose basal expression level is higher in females, would be expressed even more highly in females. This is because, in reality, Vip suppresses *fshb* expression in a female-specific manner. In addition, the lack of either Pacap or Vip would result in increased expression of both *sl* and *prl* in only one sex, leading to undesirable sex differences in the expression of genes that are equally important in both sexes. Hence, the sex-dependent alternative use of Pacap and Vip can be regarded as an excellent system that has evolved to ensure stable pituitary hormone expression regardless of circulating estrogen levels. In this sense, we can find a definite evolutionary significance in that system.

Here, we would also like to mention that it is becoming increasingly accepted that not all sex differences in the brain function to cause sex differences in physiological and behavioral traits and that some of them function to reduce sex differences caused by other factors (*e.g.*, De Vries, 2004, *Endocrinology*,

145:1063–1068; Arnold, 2014, *Exp Neurol*, 259:2–9; McCarthy, 2016, *Philos Transact Royal Soc B Biol Sci*, 371:20150106). The sex difference in Pacap and Vip expression identified in the present study is considered a typical example of such sex differences. We hope that the reviewer agrees that not only sex differences that produce different phenotypes in males and females but also sex differences that reduce existing sex differences, have physiological and evolutionary significance.

However, we agree with the reviewer that the manuscript lacked discussion from an evolutionary perspective. Therefore, we have made the following revisions to the manuscript, based on the above arguments:

Line 427: The summary sentence “This differential use of neuropeptides for brain–pituitary signaling most probably functions to prevent sex differences in pituitary hormones, by compensating for differences in the steroid milieu.” has been modified and now reads “It is becoming increasingly accepted that not all sex differences in the brain cause sex differences in physiological and behavioral traits and that some of them serve to prevent sex differences in these traits (De Vries, 2004; Arnold, 2014; McCarthy, 2016). The differential expression of neuropeptides for brain–pituitary signaling identified in the present study is considered a typical example of such sex differences and most probably serves to prevent undesirable sex differences in gametogenesis, ion and water transport, and skin coloration. The differential use of neuropeptides that are oppositely regulated by estrogen may represent a regulatory mechanism that has evolved to ensure these pituitary functions regardless of the estrogen milieu.”.

Line 427: With the above change, “sex steroid milieu” has been replaced with “estrogen milieu”.

Reference list: The following two references, which are cited in the above text, have been added.

De Vries GJ (2004) Sex differences in adult and developing brains: compensation, compensation, compensation. *Endocrinology* 145:1063–1068. doi:10.1210/en.2003-1504.

Arnold AP (2014) Conceptual frameworks and mouse models for studying sex differences in physiology and disease: why compensation changes the game. *Exp Neurol* 259:2–9. doi:10.1016/j.expneurol.2014.01.021.

Line 395: In addition, in response to this comment and reviewer #2’s specific comment 1, we have added a discussion from functional and evolutionary perspectives to the Discussion section. More specifically, “Pomc is post-translationally processed in the intermediate lobe to yield α -melanocyte-stimulating hormone, whose primary function in teleosts is to regulate skin coloration (Takahashi *et al.*, 2009). Although this function is important regardless of sex, the expression of *pomc* is male-biased (this study and Kawabata-Sakata *et al.*, 2020). Similar to *sl*, the male-dominant action of Pacap may serve to reduce the sex difference in *pomc* expression, which would otherwise be even larger.” has been replaced with “Pomc is post-translationally processed in the PI to yield α -melanocyte-stimulating hormone, whose primary function in teleosts is to regulate skin coloration (Takahashi *et al.*, 2009). Consistent with the fact that in medaka, males generally have darker and more vivid skin color than females, the basal expression of *pomc* is male-biased (this study and Kawabata-Sakata *et al.*, 2020). Similar to *sl*, the male-dominant action of Pacap may serve to reduce the sex difference in *pomc* expression, which would otherwise be even larger. In other words, Pacap may prevent extreme skin darkening in males by suppressing *pomc*. Medaka prefer mates with skin color similar to their own and avoid mates with skin color that differ greatly from their own, suggesting that pre-mating reproductive isolation is occurring

between strains of medaka with different skin colors (Ikawa *et al.*, 2017). The differential use of Pacap and Vip may contribute to this process by reducing sex differences in skin color through the regulation of *sl* and *pomc*.”.

Reference list: The following reference, which is newly cited in the above text, has been added.

Ikawa M, Ohya E, Shimada H, Kamijo M, Fukamachi S (2017) Establishment and maintenance of sexual preferences that cause a reproductive isolation between medaka strains in close association. *Biol Open* 6:244–251. doi:10.1242/bio.022285.

Reviewer #3’s specific comment 1:

Line 189: Can the authors quantify the percentage of cells expressing each estrogen receptor?

Response to reviewer #3’s specific comment 1:

Because only a few micrographs were taken, it was not possible to calculate the percentage of neurons co-expressing estrogen receptors. Therefore, we have again performed double-label *in situ* hybridization for *adcyp1/vip* and *esr1/esr2a/esr2b* with a larger number of samples and taken many more micrographs to calculate the percentage of neurons co-expressing estrogen receptors. The results showed that both *adcyp1*- and *vip*-expressing PMp neurons predominantly express *esr1* and *esr2b*, as shown below (the new Figure 5A; the graph on the right shows the data for *adcyp1*-expressing neurons, and the graph on the left shows the data for *vip*-expressing neurons).

To include these results, we have revised the manuscript as follows.

Line 191: “Hence, estrogen can potentially act directly, at least partially, on *adcyp1*- and *vip*-expressing neurons, and the action of estrogen may be mediated by all three subtypes of ER.” has been altered to read “In both *adcyp1*- and *vip*-expressing neurons, the percentage of neurons co-expressing *esr1* and *esr2b* was significantly higher compared with that of neurons co-expressing *esr2a* in both sexes ($p = 0.0071$ and 0.0003 for *esr1* and *esr2b*, respectively, in male *adcyp1* neurons; $p = 0.0002$ and < 0.0001 for *esr1* and *esr2b*, respectively, in female *adcyp1* neurons; $p = 0.0016$ and 0.0004 for *esr1* and *esr2b*, respectively, in male *vip* neurons; $p < 0.0001$ for both *esr1* and *esr2b* in female *vip* neurons) (*adcyp1* neurons: main effect of sex, $p = 0.0012$; main effect of ER subtype, $p < 0.0001$; interaction between sex and ER subtype, $p = 0.1477$. *vip* neurons: main effect of sex, $p = 0.0248$; main effect of ER subtype, $p < 0.0001$; interaction between sex and ER subtype, $p = 0.0638$) (Figure 5A). Hence, estrogen can potentially act directly, at least partially, on *adcyp1*- and *vip*-expressing neurons, and the action of estrogen may be mediated predominantly by *esr1* and *esr2b*.”.

Line 343: “the Pacap- and Vip-expressing neurons in the POA express all three ER subtypes” has been changed to “the Pacap- and Vip-expressing neurons in the POA express all three ER subtypes, with *Esr1* and *Esr2b* being predominant”.

Line 547: “Double-label *in situ* hybridization was done as described by Kawabata-Sakata *et al.* (2020). In brief, the brain was fixed in 4% PFA and embedded in 5% agarose (Type IX-A; Sigma-Aldrich) supplemented with 20% sucrose. Frozen coronal sections of 20- μ m thickness were cut and hybridized simultaneously with the above-mentioned *adcyp1* or *vip* probe, which was labeled with fluorescein by using Fluorescein RNA Labeling Mix and T7 RNA polymerase (Roche Diagnostics), and a DIG-labeled *vip* or ER (*esr1*, XM_020714493; *esr2a*, NM_001104702; *esr2b*, NM_001128512) probe described previously (Hiraki *et al.*, 2012). The fluorescein-labeled probe was visualized by using horseradish peroxidase-conjugated anti-fluorescein antibody and the TSA Plus Fluorescein System (PerkinElmer, Waltham, MA); the DIG-labeled probe was visualized by using alkaline phosphatase-conjugated anti-DIG antibody and Fast Red (Roche Diagnostics). Cell nuclei were counterstained with 4',6-diamidino-2-phenylindole (DAPI). Fluorescent images were acquired by using a confocal laser scanning microscope (Leica TCS SP8; Leica Microsystems, Wetzlar, Germany). The following excitation and emission wavelengths were used for detection: DAPI, 405 nm and 410–480 nm, respectively; fluorescein, 488 nm and 495–545 nm, respectively; and Fast Red, 552 nm and 620–700 nm, respectively.” has been modified and now reads “Double-label *in situ* hybridization was essentially done as described by Kawabata-Sakata *et al.* (2020). In brief, the brain and pituitary were fixed in 4% PFA and embedded in paraffin (for calculation of the percentage of *adcyp1/vip*-expressing neurons that co-express ER genes and co-expression analysis of Pacap/Vip receptor and pituitary hormone genes) or 5% agarose (Type IX-A; Sigma-Aldrich) supplemented with 20% sucrose for cryoprotection (for co-expression analysis of *adcyp1/vip* and ER genes and that of *adcyp1* and *vip*). Paraffin sections of 10- μ m thickness or frozen sections of 20- μ m thickness were cut in the coronal plane and hybridized simultaneously with fluorescein- and DIG-labeled cRNA probes. The *adcyp1*, *vip*, and Pacap/Vip receptor probes mentioned above and the ER (*esr1*, XM_020714493; *esr2a*, NM_001104702; *esr2b*, NM_001128512) probes described previously (Hiraki *et al.*, 2012) were labeled with fluorescein by using Fluorescein RNA Labeling Mix and T7 RNA polymerase (Roche Diagnostics). The above-mentioned *vip* and ER probes and the pituitary hormone (*fshb*, NM_001309017; *lhb*, NM_001137653; *tshb*, XM_004068796; *gh*, XM_004084500; *sl*, NM_001104790; *prl*, XM_004071867; *pomc*, XM_004066456) probes described previously (Kawabata-Sakata *et al.*, 2020) were labeled with DIG as described above. The fluorescein-labeled probe was visualized by using horseradish peroxidase-conjugated anti-fluorescein antibody and the TSA Plus Fluorescein System (PerkinElmer, Waltham, MA, USA); the DIG-labeled probe was visualized by using anti-DIG mouse primary antibody (Abcam, Cambridge, UK) and Alexa Fluor 555-conjugated goat anti-mouse IgG secondary antibody (Thermo Fisher Scientific) (for calculation of the percentage of *adcyp1/vip*-expressing neurons that co-express ER genes and co-expression analysis of Pacap/Vip receptor and pituitary hormone genes) or by using alkaline phosphatase-conjugated anti-DIG antibody and Fast Red (Roche Diagnostics) (for co-expression analysis of *adcyp1/vip* and ER genes and that of *adcyp1* and *vip*). Cell nuclei were counterstained with 4',6-diamidino-2-phenylindole (DAPI). Fluorescent images were acquired by using a confocal laser scanning microscope (Leica TCS SP8; Leica Microsystems, Wetzlar, Germany). The following excitation and emission wavelengths were used for detection: DAPI, 405 nm and 410–480 nm, respectively; fluorescein, 488 nm and 495–545 nm, respectively; and Fast Red and Alexa Fluor 555, 552 nm and 620–700 nm, respectively. In the calculation of the percentage of *adcyp1*- and *vip*-neurons

that co-express ER, four or five PMp sections per individual were photographed and examined for ER expression in all *adcyp1*- and *vip*-neurons in the sections (a total of more than 30 neurons for each individual).”.

Line 644: “Two-way ANOVA followed by either Bonferroni’s (for comparisons among experimental groups) or Dunnett’s (for comparisons of experimental versus control groups) *post hoc* test was used for analyses of *adcyp1/vip* expression throughout the diurnal/reproductive cycle and during growth/sexual maturation and pituitary hormone gene expression in response to Pacap/Vip supplementation.” has been replaced with “Two-way ANOVA followed by either Bonferroni’s (for comparisons among experimental groups) or Dunnett’s (for comparisons of experimental versus control groups) *post hoc* test was used for analyses of *adcyp1/vip* expression throughout the diurnal/reproductive cycle and during growth/sexual maturation, ER expression in *adcyp1*- and *vip*-neurons, and pituitary hormone gene expression in response to Pacap/Vip supplementation.”.

Figure 5: The graphs shown above have been included in the revised version of the paper as Figure 5A (these images have been added to the original Figure 4, but this caused the size of the figure to exceed one page; accordingly, we have divided it into two figures (new Figures 5 and 6)).

Line 908: The legend for the new figure 5 reads: “*adcyp1*- and *vip*-expressing neurons in the POA are direct targets of estrogen. (A) Percentage of *adcyp1*- and *vip*-expressing neurons in the male and female PMp that co-express estrogen receptors (*esr1*, *esr2a*, and *esr2b*) (n = 3 fish per group). (B, C) Representative micrographs showing the expression of estrogen receptors in *adcyp1*-expressing (B) and *vip*-expressing (C) neurons in the male and female PMp. In each row, left and middle panels show images of, respectively, *adcyp1/vip* (green) and estrogen receptor (magenta) expression in the same section; right panel shows the merged image with nuclear counterstaining (blue). Arrowheads indicate representative *adcyp1/vip*-expressing neurons that co-express estrogen receptor. Scale bars represent 20 μ m. Statistical differences were assessed by Bonferroni’s *post hoc* test (A). ***p* < 0.01; ****p* < 0.001.”.

Lines 190 and 191: With this change, “Figure 4A” and “Figure 4B” have been changed to “Figure 5A and B” and “Figure 5A and C”, respectively.

Reviewer #3’s specific comment 2:

Line 203: What percentage of the cells were labeled by the neurobiotin? Were only PMp neurons labeled by the neurobiotin or did the authors observe retrograde labeling in PACAP/VIP neurons located in other areas?

Response to reviewer #3’s specific comment 2:

Because the percentage of neurons labeled with the tracer dye varies greatly depending on the parameters of the experiment, calculating the percentage is not that meaningful. However, in response to reviewer #2’s specific comment 3 and reviewer #3’s specific comment 11, we have performed immunohistochemical analysis of Pacap and Vip in the anterior pituitary and observed a dense plexus of Pacap- and Vip-containing axons, suggesting that a significant percentage of neurons project to the anterior pituitary. Please see Response to reviewer #2’s specific comment 3 for further details.

We did not check whether Pacap and Vip neurons located outside the PMp were also labeled with

Neurobiotin. However, of the brain nuclei in which these peptidergic neurons reside, only the PMp is known to project to the anterior pituitary. Therefore, it is likely that Pacap and Vip neurons outside of the PMp do not project to the anterior pituitary.

Reviewer #3's specific comment 3:

Line 211: The word "primarily" means that most of the neurons project to the pituitary. Is that the case?

Response to reviewer #3's specific comment 3:

Line 222: We have removed the word "primarily" since it is still uncertain whether the majority of those neurons project to the pituitary.

Reviewer #3's specific comment 4:

Lines 237-341: Since the authors do not co-localize receptors onto specific cell types in the pituitary, they should at least try to match the receptor expression patterns to the known expression locations of pituitary hormone cells.

Response to reviewer #3's specific comment 4:

In response to this comment, we have performed double-label *in situ* hybridization for the respective Pacap/Vip receptors with pituitary hormones to see in which hormone-producing cells of the pituitary these receptors are expressed. The results showed that *adcyp1r1a* and *vipr2b* are expressed in *pomc*-expressing cells in the rostral pars distalis (RPD) and pars intermedia (PI), respectively, and *adcyp1r1b* is expressed in *sl*-expressing cells, which are located in the PI, as shown below (the new Figure 7F; representative micrographs showing the presence or absence of co-expression of the respective Pacap/Vip receptors and pituitary hormones in the pituitary. Arrowheads mark representative cells co-expressing the indicated receptor and pituitary hormone, while arrows mark the receptor-expressing cells that do not express the indicated pituitary hormone. Asterisks denote non-specific labeling of blood vessels).

This new information has provided major insights into the mechanisms of pituitary hormone regulation by Pacap and Vip, and we have extensively revised the manuscript accordingly as follows:

Line 248: "*adcyp1r1a* and *adcyp1r1b* were expressed in the anterior lobe of the pituitary, whereas

adcyp1r1b and *vipr2b* were expressed in the intermediate lobe (Figure 5C)” has been replaced with “*adcyp1r1a* was expressed in the rostral pars distalis (RPD) of the pituitary, whereas *adcyp1r1b* and *vipr2b* were expressed in the pars intermedia (PI) (Figure 7C)”.

Line 252: “We further investigated the identity of the pituitary cells expressing these receptors. The RPD of teleosts, including medaka, contains prolactin (*prl*)- and pro-opiomelanocortin (*pomc*)-expressing cells, whereas the PI contains somatolactin (*sl*)- and *pomc*-expressing cells (Royan *et al.*, 2021). Double *in situ* hybridization for the receptors and pituitary hormone genes showed that *adcyp1r1a* was expressed in *pomc*-, but not *prl*-expressing, cells in the RPD, and *adcyp1r1b* and *vipr2b* were exclusively expressed in *sl*- and *pomc*-expressing cells, respectively, in the PI (Figure 7F).” has been added.

Reference list: The following reference, which is cited in the above text, has been added.

Royan MR, Siddique K, Csucs G, Puchades MA, Nourizadeh-Lillabadi R, Bjaalie JG, Henkel CV, Weltzien FA, Fontaine R (2021) 3D atlas of the pituitary gland of the model fish medaka. bioRxiv doi:10.1101/2021.05.31.446412.

Line 366: “In grass carp (*Ctenopharyngodon idellus*) and goldfish (*Carassius auratus*), Pacap has been shown to act directly on SI-producing cells and to stimulate the synthesis and release of SI via Adcyp1r1 (Jiang *et al.*, 2008; Azuma *et al.*, 2009; 2013). Taken together with our finding that *adcyp1r1b*, but not *adcyp1r1a*, is expressed in the intermediate lobe where SI-producing cells reside, Adcyp1r1b most probably mediates the stimulatory effects on *sl* expression.” has been modified and now reads “Combining this with our additional finding that *sl*-expressing cells express *adcyp1r1b*, Pacap and Vip may act directly on these cells to stimulate *sl* expression via Adcyp1r1b. Consistent with this notion, in grass carp (*Ctenopharyngodon idellus*) and goldfish (*Carassius auratus*), Pacap has been shown to act directly on SI-producing cells and to stimulate the synthesis and release of SI via Adcyp1r1 (Jiang *et al.*, 2008; Azuma *et al.*, 2009; 2013). In contrast, since no expression of Pacap/Vip receptors was detected in *prl*-expressing cells, it seems unlikely that Pacap and Vip have direct effects on *prl* expression. Considering that *pomc*-expressing cells in the RPD adjacent to *prl* cells express *adcyp1r1a*, Pacap and Vip may stimulate *prl* expression through their action on these *pomc* cells.”.

Line 389: “the inhibitory effects of Pacap and Vip on *pomc* were evident only in male pituitaries and thus seem to be mediated by *vipr2b*, which is expressed more abundantly in male than in female pituitaries. Given the distribution of *vipr2b* expression in the intermediate lobe, where *pomc*-expressing cells reside, Pacap and Vip administered to pituitary cultures may have acted directly on *pomc*-producing cells to repress the expression of *pomc*. It is plausible that, *in vivo*, Pacap primarily serves this function in a male-specific manner. Because *adcyp1r1b* is also expressed in the intermediate lobe and is activated more potently by Pacap than by Vip, Adcyp1r1b in addition to Vipr2b may possibly contribute to the male-specific repression of *pomc* by Pacap *in vivo*.” has been changed to “the inhibitory effects of Pacap on *pomc* were evident only in male pituitaries and thus seem to be mediated by *vipr2b*, which is expressed more abundantly in male than in female pituitaries. Given the expression of *vipr2b* by *pomc*-expressing cells, Pacap may act directly on these cells to repress the expression of *pomc*.”.

Line 416: “(although we were unable to detect this expression)” has been added.

Line 547: The description of double-label *in situ* hybridization in the Methods section has been revised.

Please see Response to reviewer #3's specific comment 1 for further details.

Figure 7: The micrographs shown above have been included in the revised version of the paper as Figure 7F.

Line 947: "(F) Identity of the pituitary cells expressing each Pacap/Vip receptor. The spatial expression pattern of each receptor was compared with that of each pituitary hormone gene in the RPD and PI. Left and middle panels show images of the expression of the indicated receptor (*adcyp1r1a*, *adcyp1r1b*, or *vipr2b*; green) and pituitary hormone gene (*sl*, *prl*, or *pomc*; magenta) in the same sections; right panels show the merged images with nuclear counterstaining (blue). Arrowheads mark representative cells co-expressing the indicated receptor and pituitary hormone gene, while arrows mark the receptor-expressing cells that do not express the indicated pituitary hormone gene. Asterisks denote non-specific labeling of blood vessels. Scale bars represent 20 μm ." has been added to the legend in Figure 7.

Reviewer #3's specific comment 5:

Line 267: reconsider the term hypophysiotropic functions. So far in the results the authors have shown hypophysiotropic projections and expression of receptors. Functions is the next stage.

Response to reviewer #3's specific comment 5:

Line 284: We agree with this comment and have replaced "Pacap and Vip produced in the PMp have hypophysiotropic functions" with "Pacap and Vip produced in the PMp are transported to the anterior pituitary, where their cognate receptors are expressed".

Reviewer #3's specific comment 6:

Lines 265-294: Do receptor locations match with the hormones most affected. For example are there receptors (which ones?) in the RPD, where prolactin cells are located?

Response to reviewer #3's specific comment 6:

In response to the specific comment 4 from this reviewer, we have performed double-label *in situ* hybridization for the respective Pacap/Vip receptors with pituitary hormones to see in which hormone-producing cells of the pituitary these receptors are expressed. The results showed that *adcyp1r1a* and *vipr2b* are expressed in *pomc*-expressing cells in the RPD and PI, respectively, and *adcyp1r1b* is expressed in *sl*-expressing cells, suggesting that the effects of Pacap and Vip on these pituitary hormones are direct, while their effects on other hormones are indirect. We have included such discussions in the Discussion section. Please see Response to reviewer #3's specific comment 4 for further details.

Reviewer #3's specific comment 7:

Lines 265-294: Considering the much stronger expression of *vip2* receptor in males, one would expect that the effect on male pituitaries (at least for hormones expressed in the posterior part) would be much stronger. There does not seem to be such a response.

Response to reviewer #3's specific comment 7:

As this reviewer notes, the male-biased expression of *vipr2b* in the pituitary would predict a male-biased effect of Pacap or Vip on some pituitary hormone. In fact, we found that Pacap has an inhibitory effect on *pomc* expression only in males. In addition, in response to the specific comment 4 from this reviewer, we have performed double-label *in situ* hybridization for Pacap/Vip receptors and pituitary hormones

and found that *vipr2b* is expressed in *pomc*-expressing cells.

Line 389: Based on these results, we have included the following text in the Discussion section: “the inhibitory effects of Pacap on *pomc* were evident only in male pituitaries and thus seem to be mediated by *vipr2b*, which is expressed more abundantly in male than in female pituitaries. Given the expression of *vipr2b* by *pomc*-expressing cells, Pacap may act directly on these cells to repress the expression of *pomc*.”.

Reviewer #3’s specific comment 8:

Lines 265-294: Why should the same peptides have stimulatory effects on the expression of some hormones and inhibitory effects on others? Does this match known signal-transduction cascades of the different receptors?

Response to reviewer #3’s specific comment 8:

All subtypes of Pacap and Vip receptors, when activated, increase intracellular cAMP levels (please see Figure 7B), suggesting that they essentially have facilitatory effects on gene expression. However, as the reviewer points out, Pacap and Vip inhibit the expression of *fshb* and *pomc*. Although no information is available for the inhibitory effect on *pomc* other than our present study, there have been several published reports on *fshb*. Notably, it has been shown in rats that PACAP is effective in inhibiting *Fshb* expression even when administered to dispersed pituitary cells, indicating a direct action of PACAP on gonadotrophs (Köves *et al.*, 2020, *Front Endocrinol*, 11:88; Winters and Moore, 2020, *Mol Cell Endocrinol* 518:110912). Furthermore, this effect has been shown to be mediated by increased expression of follistatin. It thus seems possible that Pacap may have a similar mechanism of action in medaka.

Line 407: To provide this information, we have added the following text to the Discussion section: “Studies on dispersed rat pituitary cells indicate that PACAP acts directly on gonadotrophs to inhibit *Fshb* expression and that this effect is mediated by increased expression of follistatin (Köves *et al.*, 2020; Winters and Moore, 2020). Possibly, Pacap may have a similar mechanism of action in medaka.”.

Line 411: With the addition of this text, “this effect” has been replaced with “the inhibitory effect on *fshb*”.

Reviewer #3’s specific comment 9:

Line 305: Are there other examples of males and females using different neuropeptides for equivalent signaling functions?

Response to reviewer #3’s specific comment 9:

To the best of our knowledge, this is the first report showing the use of different neuropeptides in males and females for equivalent signaling functions. We stated “this is the first study to report...” in the previous sentence (line 319), and therefore did not repeat it in this sentence.

Reviewer #3’s specific comment 10:

Line 308: Is it possible that these peptides are co-expressed with other hypophysiotropic peptides with more established pituitary functions?

Response to reviewer #3's specific comment 10:

The best known and most well-studied hypophysiotropic peptide in teleosts is probably gonadotropin-releasing hormone 1 (Gnrh1). In medaka, the hypophysiotropic population of Gnrh1 neurons reside not in the PMp, but in the nucleus located laterally to it (Kawabata *et al.*, 2012, *Neuroscience*, 218:65–77; Karigo *et al.*, 2012, *Endocrinology*, 153:3394–3404), and thus are clearly different from Pacap and Vip neurons in the PMp. Further investigation may reveal neuropeptides that are co-expressed with Pacap and Vip in the PMp, but so far, no such peptides have been identified.

Reviewer #3's specific comment 11:

Line 395: As mentioned earlier, the conclusion does not make much evolutionary sense. Is it possible that considering the direct innervation of the pituitary, the neurons target different pituitary cells in males and females? Or that neurons from this nucleus target other brain areas which differentially express the receptors? Or that the different affinities of the receptors for the ligands induce a differential response?

Response to reviewer #3's specific comment 11:

Line 215: In response to this comment and reviewer #2's specific comment 3, we have performed immunohistochemical analysis of Pacap and Vip and observed a dense plexus of Pacap- and Vip-containing axons throughout the anterior pituitary in both males and females (please see Response to reviewer #2's specific comment 3 for further details). Because there were no noticeable sex differences in the distribution of axons, it seems unlikely that Pacap and Vip neurons target different pituitary cells in females and males. To include this information in the revised manuscript, the following text has been added: **“There were no noticeable differences in the distribution of axons between the sexes.”**

In this immunohistochemical analysis, we also tried to determine whether Pacap and Vip neurons in the PMp project to other brain regions; however, we could not observe any distinct immunoreactive axons originating from the PMp except in the pituitary. Therefore, in order to clarify this, it is necessary to perform further analyses, such as generating GFP transgenic fish and imaging the axonal projection patterns of these neurons in detail, which will be the subject of future work.

As this reviewer points out, affinity to Pacap and Vip differs among the receptors, but all of them commonly increase intracellular cAMP levels when activated. Therefore, it is unlikely that different receptors induce different responses.

Reviewer #3's specific comment 12:

Lines 454-462: Only a single reference gene was used for all real-time quantifications? Can the authors show that its levels were not affected by treatments?

Response to reviewer #3's specific comment 12:

In the present study, only *actb* was used for normalization of real-time PCR results. Our previous real-time PCR studies performed on medaka brains, with and without hormone treatment, have shown that normalization to *actb* gave the identical results as that to other housekeeping genes such as glyceraldehyde-3-phosphate dehydrogenase (*gapdh*) and ribosomal protein L13 (*rpl13*). This indicates that *actb* is an appropriate reference gene in medaka brains (Yamashita *et al.*, 2017, *J Neuroendocrinol*, doi:10.1111/jne.12545).

In contrast, we have not verified which gene is most appropriate as a reference gene in hormone-treated

medaka pituitaries. In the present study, the entire amount of total RNA extracted from the cultured pituitaries was reverse-transcribed without quantification, and the amount of reverse-transcribed products was supposed to vary among samples. In fact, the expression levels of *actb* actually varied among samples, and it is thus impossible to show that they are unaffected by peptide treatment. Accordingly, here we chose to measure the expression of two additional housekeeping genes (*gapdh* and *rpl13*) and normalize the results to the geometric mean of a total of three genes (*actb*, *gapdh*, and *rpl13*) according to Vandesompele *et al.* (2002, *Genome Biol*, 3:0034). Reanalysis with this normalization yielded results that were almost identical to those originally obtained with *actb* alone. The original results (previous Figure 6) and the new results obtained with this normalization (new Figure 8) are shown below.

The original results (previous Figure 6)

The new results obtained with this normalization (new Figure 8)

However, out of a total of 28 between-group comparisons, a significant difference was additionally detected in one comparison (*lhb* expression in female pituitaries, vehicle versus Pacap) and disappeared in another (*pomc* expression in male pituitaries, vehicle versus Vip). The Results, Discussion, and Methods sections were thus revised accordingly, as follows. These revisions did not affect the conclusions of the paper and did not require any revisions to the abstract.

Line 286: “Real-time PCR revealed that expression of the follicle-stimulating hormone (Fsh) β subunit (*fshb*) gene was significantly reduced by both Pacap and Vip supplementation in cultured female pituitaries ($p = 0.0027$ and 0.0241 , respectively), but not in male pituitaries (main effect of sex, $p < 0.0001$; main effect of treatment, $p = 0.0411$; interaction between sex and treatment, $p = 0.0343$) (Figure 6). Conversely, expression of the luteinizing hormone (Lh) β subunit (*lhb*) gene in male pituitaries was increased by Pacap supplementation ($p = 0.0228$), whereas Vip had no significant effect (main effect of sex, $p = 0.5581$; main effect of treatment, $p = 0.0004$; interaction between sex and treatment, $p = 0.7072$) (Figure 6). A similar trend in *lhb* expression was observed in female pituitaries, although the effect of Pacap did not reach significance ($p = 0.0975$). Pacap and Vip had no significant effects on expression of the thyroid-stimulating hormone β subunit (*tshb*) or growth hormone (*gh*) gene in either sex (*tshb*: main effect of sex, $p = 0.7380$; main effect of treatment, $p = 0.4348$; interaction between sex and treatment, $p = 0.2138$. *gh*: main effect of sex, $p < 0.0001$; main effect of treatment, $p = 0.4576$; interaction between sex and treatment, $p = 0.2555$) (Figure 6). The levels of somatotactin (*sl*) and prolactin (*prl*) expression were substantially increased by both Pacap and Vip supplementation in both sexes (*sl*: $p = 0.0170$, 0.0002 , <0.0001 , and <0.0001 for Pacap in males, Vip in males, Pacap in females, and Vip in females, respectively; main effect of sex, $p < 0.0001$; main effect of treatment, $p < 0.0001$; interaction between sex and treatment, $p = 0.0098$. *prl*: $p = 0.0021$, 0.0005 , <0.0001 , and <0.0001 for Pacap in males, Vip in males, Pacap in females, and Vip in females, respectively; main effect of sex, $p = 0.0096$; main effect of treatment, $p < 0.0001$; interaction between sex and treatment, $p = 0.4376$) (Figure 6). The pro-opiomelanocortin (*pomc*) gene expression was reduced by both Pacap and Vip

supplementation in male pituitaries ($p = 0.0002$ and 0.0071 , respectively), but showed no significant change in female pituitaries (main effect of sex, $p < 0.0001$; main effect of treatment, $p = 0.0182$; interaction between sex and treatment, $p = 0.0057$) (Figure 6).” has been modified and now reads “Real-time PCR revealed that expression of the follicle-stimulating hormone (Fsh) β subunit (*fshb*) gene was significantly reduced by both Pacap and Vip supplementation in cultured female pituitaries ($p = 0.0176$ and 0.0445 , respectively), but not in male pituitaries (main effect of sex, $p < 0.0001$; main effect of treatment, $p = 0.1252$; interaction between sex and treatment, $p = 0.0557$) (Figure 8). Expression of the luteinizing hormone (Lh) β subunit (*lhb*) gene in both male and female pituitaries was increased by Pacap supplementation ($p = 0.0045$ and 0.0452 , respectively), whereas Vip had no significant effect (main effect of sex, $p = 0.0743$; main effect of treatment, $p < 0.0001$; interaction between sex and treatment, $p = 0.6211$) (Figure 8). Pacap and Vip had no significant effects on expression of the thyroid-stimulating hormone β subunit (*tshb*) or growth hormone (*gh*) gene in either sex (*tshb*: main effect of sex, $p = 0.6452$; main effect of treatment, $p = 0.5120$; interaction between sex and treatment, $p = 0.3104$. *gh*: main effect of sex, $p < 0.0001$; main effect of treatment, $p = 0.8933$; interaction between sex and treatment, $p = 0.1613$) (Figure 8). The levels of *sl* and *prl* expression were substantially increased by both Pacap and Vip supplementation in both sexes (*sl*: $p = 0.0079$ for Pacap in males and <0.0001 for Pacap in females and Vip in both sexes; main effect of sex, $p < 0.0001$; main effect of treatment, $p < 0.0001$; interaction between sex and treatment, $p = 0.0108$. *prl*: $p = 0.0005$ for Pacap in males and <0.0001 for Pacap in females and Vip in both sexes; main effect of sex, $p = 0.1199$; main effect of treatment, $p < 0.0001$; interaction between sex and treatment, $p = 0.7814$) (Figure 8). The expression of *pomc* in male pituitaries was reduced by Pacap supplementation ($p = 0.0025$), whereas Vip had no significant effect ($p = 0.0617$). In contrast, neither Pacap nor Vip had no significant effects on *pomc* expression in female pituitaries (main effect of sex, $p < 0.0001$; main effect of treatment, $p = 0.1068$; interaction between sex and treatment, $p = 0.0239$) (Figure 8).”

Line 308: “both Pacap and Vip have sex-independent stimulatory effects on the expression of *sl* and *prl*, and sex-dependent inhibitory effects on *fshb* and *pomc*. In addition, Pacap exerts potentially sex-independent stimulatory effects on *lhb*.” has been changed to “both Pacap and Vip have sex-independent stimulatory effects on the expression of *sl* and *prl*, and female-specific inhibitory effects on *fshb*. In addition, Pacap exerts sex-independent stimulatory effects on *lhb* and male-specific inhibitory effects on *pomc*.”.

Line 355: “Pacap and Vip stimulate the expression of *sl* and *prl* in the pituitaries of both sexes, and suppress the expression of *fshb* and *pomc* in only female and male pituitaries, respectively. Pacap was additionally found to stimulate *lhb* in both sexes, albeit not statistically significant in females.” has been changed to “Pacap and Vip stimulate the expression of *sl* and *prl* in the pituitaries of both sexes and suppress the expression of *fshb* only in female pituitaries. Pacap was additionally found to stimulate *lhb* in both sexes and suppress *pomc* only in males.”.

Line 389: “the inhibitory effects of Pacap and Vip on *pomc* were evident only in male pituitaries and thus seem to be mediated by *vipr2b*, which is expressed more abundantly in male than in female pituitaries. Given the distribution of *vipr2b* expression in the intermediate lobe, where *pomc*-expressing cells reside, Pacap and Vip administered to pituitary cultures may have acted directly on *pomc*-producing cells to repress the expression of *pomc*. It is plausible that, *in vivo*, Pacap primarily serves this function in a male-specific manner. Because *adcyap1r1b* is also expressed in the intermediate lobe and is activated more potently by Pacap than by Vip, *Adcyap1r1b* in addition to *Vipr2b* may possibly

contribute to the male-specific repression of *pomc* by Pacap *in vivo*.” has been changed to “the inhibitory effects of Pacap on *pomc* were evident only in male pituitaries and thus seem to be mediated by *vipr2b*, which is expressed more abundantly in male than in female pituitaries. Given the expression of *vipr2b* by *pomc*-expressing cells, Pacap may act directly on these cells to repress the expression of *pomc*.”.

Line 499: “The suitability of *actb* as a reference gene in the medaka brain has been demonstrated previously (Yamashita *et al.*, 2017).” has been added.

Line 626: “Melting curve analysis and data normalization were performed as described above.” has been changed to “Melting curve analysis was performed as described above. Data were normalized to the geometric mean of three reference genes, *actb*, *gapdh* (encoding glyceraldehyde-3-phosphate dehydrogenase; National BioResource Project (NBRP) Medaka clone ID olbrno9_f15), and *rpl13* (encoding ribosomal protein L13; NBRP Medaka clone ID olovano51_b24), according to Vandesompele *et al.* (2002).”.

Reference list: The following reference, which is cited in the above text, has been added.

Vandesompele J, De Preter K, Pattyn F, Poppe B, Van Roy N, De Paepe A, Speleman F (2002) Accurate normalization of real-time quantitative RT-PCR data by geometric averaging of multiple internal control genes. *Genome Biol* 3:0034. doi:10.1186/gb-2002-3-7-research0034.

Table S3: The information on primers for *gapdh* and *rpl13* has been added as follows.

Table S3. Primers used for real-time PCR.

target	direction	sequence (5′–3′)
adcyp1	forward	TGTCCTACAAGAGCTAGA
adcyp1	reverse	CGTGGAAAGAGCCAGAAA
vip	forward	TCAGACGCCATCTTACACA
vip	reverse	CAGGTA CTCTTGACTGCCATC
fshb	forward	GACTGGTCCTACGAAAGTTA
fshb	reverse	TGTGGTTCTTGTTGCATGT
lhb	forward	GTGGATCCGTCAGTCACATAACC
lhb	reverse	GTGCAGTCAGACGCGTTCAT
tshb	forward	TTACCTACCCCGTGGCACTC
tshb	reverse	TGCGTGCACTCATCACTGTC
gh	forward	CTTTTCTCTGACTTTGAGAGTT
gh	reverse	GTGCTTGTCTAATGGGCTGATG
sl	forward	GCATCACCAAAGCATTACC
sl	reverse	ATGCAGCAGCCATTTATCAGA
prl	forward	TCCTGTCCA ACTCTGCAA ACTC
prl	reverse	CAGGTTCTGGAATGCTCCT
pomc	forward	AGCAGCATGACGGAGT
pomc	reverse	GGAGAGATGAAAGAGAAGGGA
actb	forward	CCCCACCCAAAGTTTAG
actb	reverse	CAACGATGGAGGAAAGACA
gapdh	forward	GACCTCCATGTTGGAATCAATG
gapdh	reverse	AATGAAGGGGTCGTTGATGG

rpl13	forward	ACTCATCCTGTTCCCAAGGAAG
rpl13	reverse	CCACTGAGCTGAGTAGCCATCTT

Reviewer #3's specific comment 13:

Line 527: The reference states 30-60 minutes for the neurobiotin retrograde labeling. Was 5 minutes enough in the current study?

Response to reviewer #3's specific comment 13:

The incubation time was chosen to be short enough to avoid diffusion of Neurobiotin injected into the pituitary, which would result in high background and false-positive signals. A longer incubation time may have resulted in more neurons being labeled, but since some POA neurons were labeled even with a 5-minute incubation, we see no problem with this methodology.

Reviewer #3's specific comment 14:

Fig 1D: Could the authors mark the spawning hour with an arrow?

Response to reviewer #3's specific comment 14:

Figure 1D: According to this comment, the typical time of spawning has been shown in the graphs.

Line 859: With this change, the sentence “**Arrowheads indicate the typical time of spawning.**” has been added to the figure legend.

Reviewer #3's specific comment 15:

Fig 1E: Could the authors mark age of puberty with an arrow?

Response to reviewer #3's specific comment 15:

Figure 1E: According to this comment, the typical time of first spawning has been shown in the graphs.

Line 861: With this change, the sentence “**Arrowheads indicate the typical time of first spawning.**” has been added to the figure legend.

Reviewer #3's specific comment 16:

Line 739: I think "in" should be "on".

Response to reviewer #3's specific comment 16:

Line 857: “**in the left**” has been corrected to “**on the left**”. We thank the reviewer for noticing this error.

Reviewer #3's specific comment 17:

Fig 2: Nice anatomical maps like in the sup figs will help reader's orientation.

Response to reviewer #3's specific comment 17:

Figures 2 and 3: Following this comment, we have added line drawings of the medaka brain (the same as panels A and B of Figures S1 and S2) to Figure 2. This caused the size of Figure 2 to exceed one page; accordingly, we have divided it into two figures (new Figures 2 and 3 shown below).

Figure 2

Figure 3

Line 866: The legend for Figure 2 now reads: “Figure 2. The POA is the major source of the sex-biased expression of *adcyp1* in the brain. (A) Lateral view (anterior to the left) of the medaka brain showing the approximate levels of sections in panel B. (B) Line drawings of coronal brain sections showing the location of nuclei containing *adcyp1*-expressing neurons (stars). (C) Number of *adcyp1*-expressing neurons in each brain nucleus of males and females (n = 5 per sex). The data are split into two graphs for visual clarity. (D) Total area of *adcyp1* expression signals in DI/Dm/Dd/Dp, rHd, lHd, PGm, and LV nuclei, where *adcyp1*-expressing neurons were tightly clustered, and their exact number could not be determined (n = 5 per sex). (E) Representative micrographs showing *adcyp1* expression in nuclei where significant differences between the sexes were detected. Scale bars represent 50 μ m. Statistical differences were assessed by unpaired *t*-test with Bonferroni-Dunn correction (C, D). ***p* < 0.01; ****p* < 0.001. For abbreviations of brain nuclei, see Table S1.”.

Line 879: The legend for Figure 3 now reads: “Figure 3. The POA is the major source of the sex-biased expression of *vip* in the brain. (A) Lateral view (anterior to the left) of the medaka brain showing the approximate levels of sections in panel B. (B) Line drawings of coronal brain sections showing the location of nuclei containing *vip*-expressing neurons (stars). (C) Number of *vip*-expressing neurons in each brain nucleus of males and females (n = 5 per sex). The data are split into two graphs for clarity. (D) Total area of *vip* expression signals in the NVT/NRL nucleus, where *vip*-expressing neurons were tightly clustered, and their exact number could not be determined (n = 5 per sex). (E) Representative micrographs showing *vip* expression in nuclei where significant differences between the sexes were detected. Scale bars represent 50 μ m. Statistical differences were assessed by unpaired *t*-test with Bonferroni-Dunn correction (C, D). ***p* < 0.01; ****p* < 0.001. For abbreviations of brain nuclei, see Table S1.”.

As a result of this change, the subsequent figures have been renumbered.

Reviewer #3’s specific comment 18:

Fig 4C: Is that a male or a female? Mention at least in the legend.

Response to reviewer #3’s specific comment 18:

Line 924: According to this suggestion, the sentence “Representative images of *adcyp1*- and *vip*-expressing neurons in a male and female, respectively, are presented.” has been added to the legend for Figure 6A (the original Figure 4C).

Reviewer #3’s specific comment 19:

Fig 5C: Some annotation would be useful: RPD, PPD, PI etc., Maybe also a scheme of the medaka pituitary detailing the known location of the different hormone-expressing cells.

Response to reviewer #3’s specific comment 19:

Following this suggestion, we have annotated the images in Figure 7C (previously Figure 5C) with RPD, PPD, and PI, as shown below.

Line 955: “PI, pars intermedia; PPD, proximal pars distalis; RPD, rostral pars distalis.” has been added to the legend in Figure 7.

Line 253: During the revision process of this manuscript, an excellent paper was published that precisely described the spatial distribution of hormone-producing cells in the medaka pituitary (Royan *et al.*, 2021, *bioRxiv*, doi:10.1101/2021.05.31.446412). We have cited this paper and added the following statement: “The RPD of teleosts, including medaka, contains prolactin (*prl*)- and pro-opiomelanocortin (*pomc*)-expressing cells, whereas the PI contains somatolactin (*sl*)- and *pomc*-expressing cells (Royan *et al.*, 2021).”.

Reference list: Royan *et al.* (2021) have been added to the reference list.

Royan MR, Siddique K, Csucs G, Puchades MA, Nourizadeh-Lillabadi R, Bjaalie JG, Henkel CV, Weltzien FA, Fontaine R (2021) 3D atlas of the pituitary gland of the model fish medaka. *bioRxiv* doi:10.1101/2021.05.31.446412.

Reviewer #3’s specific comment 20:

Fig 6: Some kind of summary scheme for the whole paper (like a graphical abstract) would be useful.

Response to reviewer #3’s specific comment 20:

We thank the reviewer for this helpful suggestion. We have made the following schematic summary of the main findings of this study and included as Figure 9 in the revised manuscript.

Line 968: The legend for Figure 9 has been added. It reads: “**Figure 9. Schematic summary illustrating that males and females respectively use Pacap and Vip in the POA to control the production of anterior pituitary hormones. Estrogen secreted by the gonad in adulthood attenuates Pacap expression and, conversely, stimulates Vip expression in the hypophysiotropic population of POA neurons, thereby establishing and maintaining their opposite sexual dimorphism. Both Pacap and Vip have sex-independent stimulatory effects on expression of *prl* and *sl*, and female-specific inhibitory effects on *fshb*. Pacap additionally exerts sex-independent stimulatory effects on *lhb* and male-specific inhibitory effects on *pomc*.**”.

Line 313: In order to refer to this figure in the main text (in the Discussion section), “**We found that**” has been changed to “**A summary of the main findings of this study is presented in Figure 9. First, we found that**”.

Additional alterations

Additional alteration 1:

Line 4: We have added three new co-authors, Yuji Nishiike, Thomas Fleming, and Daichi Kayo, who have contributed to the revision of the manuscript. All authors have approved this change.

Line 16: The statement in Author contributions has been changed to “**JY and KO designed the research; JY, YN, TF, DK, and KO performed the research and analyzed/interpreted the data; and JY and KO wrote the manuscript.**”

Additional alteration 2:

Line 633: The subsection title “**Statistical analysis**” has been changed to “**Statistics and reproducibility**” according to the journal’s guideline.

Additional alteration 3:

Line 649: A “Data availability” section has been added according to the journal’s guideline. It reads:
“Data availability. The medaka *adcyap1* and *vip* cDNA sequences have been deposited in GenBank under accession numbers LC579549 and LC579550, respectively. All other data supporting the findings of this study are available within the article and its supplementary information or from the corresponding author upon reasonable request.”.

REVIEWERS' COMMENTS:

Reviewer #1 (Remarks to the Author):

Nil.

Reviewer #2 (Remarks to the Author):

Nothing

Reviewer #3 (Remarks to the Author):

I would like to commend the authors for performing such a comprehensive and non-compromising revision to the manuscript.

The authors adequately address my initial concerns regarding the functional/evolutionary significance of the dimorphism and bring convincing arguments to show that sexual dimorphic peptide secretion can act as a compensatory mechanism for reducing inter-sex differences. The subject is now well covered in the discussion and places the findings in a suitable evolutionary and physiological perspective.

The co-localization studies of the receptors on pituitary cells are also beautifully executed and bring new insights regarding the mechanistic effects of the peptides.

Finally, the summarizing figure 9 helps readers digest the large amount of data and its functional implications.

I happily recommend accepting the manuscript for publication.